# Oncological Outcomes of De-Escalation of Axillary Surgery in Breast Cancer Patients at a Referral Cancer Center in Colombia

**DOI:** 10.3390/cancers17213396

**Published:** 2025-10-22

**Authors:** Sandra Esperanza Díaz-Casas, Andres Augusto Reyes-Agudelo, Oscar Alberto Vergara-Gamarra, Ximena Briceño-Morales, Luis Guzmán-AbiSaab, Daniel Contreras-Perez, Carlos Lehmann-Mosquera, Javier Ángel-Aristizábal, Mauricio García-Mora, Carlos Duarte-Torres, Iván Mariño-Lozano, Raúl Suárez-Rodríguez, Marcela Núñez-Lemus

**Affiliations:** 1Functional Unit for Breast and Soft Tissue Tumors, Instituto Nacional de Cancerología, Bogotá 111321, Colombia; 2Fundación Universitaria de Ciencias de la Salud, Instituto Nacional de Cancerología, Bogotá 111321, Colombia; 3Office of the Deputy Director of Research, Epidemiological Surveillance, Promotion, and Prevention of Cancer, Instituto Nacional de Cancerología, Bogotá 111321, Colombia; mnunez@cancer.gov.co

**Keywords:** breast neoplasms, axilla, sentinel lymph node biopsy, lymph node excision, neoadjuvant therapy, lymphatic metastasis, overall survival, disease-free survival

## Abstract

This research assessed the effect of de-escalating axillary surgery on oncologic outcomes in breast cancer patients treated from 2013 to 2023 at the Instituto Nacional de Cancerologia (INC) in Colombia, in a real-world clinical practice setting in a middle income country. The applicability of the sentinel lymph node biopsy (SLND) was evaluated in 643 patients who were undergoing SLND as initial surgery and 144 patients who were undergoing SLND after neoadjuvant chemotherapy. The final results showed that oncological outcomes are related to biological subtypes, tumor size and histological grade.

## 1. Introduction

By 2022, an estimated 2,296,840 new cases of breast cancer were reported worldwide, with an age-standardized incidence rate of 46.8 per 100,000 inhabitants, making it the most frequently diagnosed cancer globally. That same year, there were approximately 666,103 deaths due to breast cancer, corresponding to an age-standardized mortality rate of 12.7 per 100,000 inhabitants, ranking it as the fourth leading cause of cancer-related death worldwide.

In Colombia, 17,018 new cases of breast cancer were reported, with an incidence rate of 50.7 per 100,000 inhabitants and a mortality rate of 13.3 per 100,000 inhabitants [1]. This type of cancer accounted for 27.7% of all neoplasms diagnosed in Colombian women, making it the neoplasm with the highest incidence and mortality in the country [2].

Breast cancer treatment involves multiple oncological specialties, with surgical management being a crucial part. Sentinel lymph node biopsy (SLNB) is a standardized, reproducible, and reliable technique that has become the gold standard for axillary staging and guiding adjuvant treatment [3].

SLNB has transformed axillary staging in early-stage breast cancer, greatly lowering morbidity without affecting patients’ overall survival (OS). In the 2000s, the multicenter ALMANAC study, published by Mansel et al. [4], showed that patients who underwent this procedure faced a lower risk of lymphedema (RR = 0.37, 95% CI: 0.23–0.60) (5% vs. 13%), less sensory loss (RR = 0.37, 95% CI: 0.27–0.50) (11% vs. 31%), and a better quality of life compared to those who underwent complete axillary lymph node dissection (ALND). The NSABP B-32 clinical trial [5], which included 5600 T1 and T2 patients with clinically negative axilla (cN0), showed that OS and disease-free survival (DFS) were similar between the group that had SLNB as the only intervention and the group that underwent ALND in those patients with negative sentinel node (SLN) (8-year OS: 91.8% vs. 90.3%; *p* = 0.12).

Subsequently, during the 2010s, the benefit of omitting ALND was studied in patients with positive nodes and low axillary burden. The ACOSOG Z0011 trial (2011) [6] randomized 891 patients with T1-2, cN0 tumors, whose SLNB pathology showed up to two SLNs positive for micro- or macrometastasis, without macroscopically pathological nodes or gross extranodal extension, who were treated with breast-conserving surgery (BCS) and adjuvant whole-breast radiotherapy (RT) using high tangential fields, demonstrating that in this group, omitting ALND had no effect on OS or DFS [7].

Nevertheless, recent evidence has identified subgroups of patients who could benefit from ALND despite meeting the surgical de-escalation criteria proposed in the ACOSOG Z0011 study [6]. In particular, Mamtani et al. [8] demonstrated that the presence of extranodal tumor deposits (ETDs) in axillary fat and microscopic extracapsular extension (ECE) >2 mm are associated with a higher likelihood of involvement of ≥4 non-sentinel nodes (OR 7.15; 95% CI: 4.04–12.67). However, the study did not evaluate clinical outcomes such as locoregional recurrence, DFS, or OS, so it cannot be affirmed that this higher lymph node burden necessarily indicates a worse prognosis. Therefore, although ETDs could be markers of high anatomical risk, there is no evidence that their presence negatively affects oncological outcomes, and their presence alone does not justify the use of ALND, even in patients who meet the criteria of the ACOSOG Z0011 study [8].

The IBCSG 23-01 trial (2013) [9] confirmed these findings in patients with micrometastasis. The AMAROS trial (2014) [10] showed that patients with SLNB and axillary radiotherapy have similar oncological control to ALND, with a lower risk of lymphedema, which was confirmed in the OTOASOR study (2021) [11], supporting the omission of ALND in these patients. The SENOMAC study [12] includes a higher proportion of patients with mastectomy (36%), up to two positive sentinel nodes with macrometastasis, and no restriction by extracapsular extension, demonstrating that omitting ALND did not compromise OS or locoregional control in any of these subgroups.

Similarly, sentinel lymph node biopsy after neoadjuvant chemotherapy (Post-NACT SLNB) has been examined in multiple studies. The ACOSOG Z1071 trial [13] showed a false negative rate (FNR) of 12.6% in cN1 patients, indicating that SLNB might not be reliable if fewer than two nodes are removed. Using dual labeling (dye and radiocolloid) reduced the FNR to 10.8%. The SN-FNAC study by Boileau et al. [14] reported an FNR of 8.4% in patients with initially positive nodes, which rose to 13.3% when considering micrometastatic disease, allowing ALND to be avoided in 30.3% of cases. The SENTINA study [15] reported an FNR of 14.2% in patients who converted from cN1 to ycN0, with a node detection rate of 80.1%. Finally, the GANEA 2 study [16] confirmed that omitting ALND is safe in cN0 patients with negative SLN treated with neoadjuvant chemotherapy (NACT), with an OS of 97.2% and an FNR of 11.9%. These studies support the selective use of Post-NACT SLNB in patients with initial axillary status of cN0 or cN1, who achieve axillary complete response (ycN0) after systemic therapy. To improve the SLNB detection rate and reduce false negatives, it is recommended to use techniques such as dual labeling (dye and radiocolloid) or the resection of at least three SLNs, consider surgical clipping of the lymph node, and utilize AE1/AE3 cytokeratins in pathology [16]. This method does not compromise oncological control and can significantly lower the morbidity associated with more invasive procedures, like ALND.

The primary outcome of this study was to evaluate the time to recurrence (TR), and the secondary objective was to assess overall survival. Additionally, prognostic factors associated with positive SLNB and prognostic factors associated with oncological outcomes of SLNB at the Functional Unit for Breast and Soft Tissue Tumors of the Instituto Nacional de Cancerología (UFM-INC) in Bogotá, Colombia, over a 10-year period (2013–2023). It focused on patients with early-stage I and IIA tumors who underwent SLNB as their initial surgical procedure (Upfront SLNB) and on those with locally advanced IIB and IIIA tumors (T3N1 only) treated with neoadjuvant chemotherapy, who then underwent SLNB (Post-NACT SLNB).

## 2. Materials and Methods

### 2.1. Study Design and Patient Eligibility

A retrospective, observational, analytical, historical cohort study was conducted, including patients with a confirmed diagnosis of infiltrating breast cancer registered in the prospective database of the UFM-INC from 1 September 2013, to 31 August 2023, who underwent sentinel lymph node biopsy as initial treatment (Upfront SLNB) in clinical stages I and IIA, or sentinel lymph node biopsy after neoadjuvant chemotherapy (Post-NACT SLNB) in clinical stages IIA, IIB, and IIIA (T3N1).

A search was performed in the UFM-INC database to find the clinical records of all patients who underwent SLNB during the specified period. These records were then reviewed in the SAP^®^ [SAP SE. *SAP ERP 6.0, Enhancement Package 7 and SAP NetWeaver 7.4*. Walldorf, Germany: SAP SE, 2025. Computer software. Accessed on 20 October 2025. https://www.sap.com] medical history system to identify patients who met the study inclusion criteria: women over 18 years old with histopathologically confirmed invasive breast cancer, clinical stage I to IIIA (T3N1 only), undergoing Upfront or Post-NACT SLNB plus primary tumor surgery, completing adjuvant systemic treatment (chemotherapy and/or targeted therapy and/or hormonal therapy), and receiving radiotherapy (RT) at the INC. The excluded patients were those diagnosed with ductal carcinoma in situ, those who had surgery elsewhere, patients who did not continue follow-up at UFM-INC after completing the initial clinical staging, patients in whom the SLN could not be identified during surgery and therefore received ALND, male patients, and those who received neoadjuvant hormonal therapy. Study variables covered clinical and pathological characteristics of the initial biopsy and surgical specimen, types of treatment administered, and clinical follow-up.

The UFM-INC protocols allow ALND to be omitted in patients with positive SLN when the histological report shows up to two nodes positive for micro- or macrometastasis, ECE < 2 mm, provided they receive adjuvant RT with high tangential fields of the whole breast or the chest wall and lower axillary region [7]. In turn, for the performance of Post-NACT SLNB, those patients are included, who have T1, T2, and T3 infiltrating breast cancer, with N0 or N1 axilla, ultrasound and clinical response in the axilla after NACT, clip placement in the N1, dual labeling (lymphoscintigraphy and methylene blue), resection of at least three nodes, lymph node processing with 2 mm thick sections for each sentinel node, and immunohistochemical studies with AE1/AE3 cytokeratin studies for all resected nodes. As a standard practice, if the SLN pathology report is positive or if the SLN is not identified during the surgical procedure, ALND is performed [17].

NACT was administered based on the biological type of breast cancer. All patients with HER2-positive tumors received anti-HER2 therapy with trastuzumab and/or pertuzumab. Adjuvant RT was ordered for all patients undergoing conservative surgery, as well as for those undergoing mastectomy with tumors larger than 5 cm, positive or close margins, and positive lymph nodes. Adjuvant systemic treatment included anti-HER2 therapy for patients with HER2-positive tumors, along with chemotherapy and endocrine therapy for those with hormone receptor (HR)-positive tumors.

Two authors extracted and entered data into an electronic platform (REDCap 7.1.2^®^) designed to record information on the clinical variables of the study. The quality and accuracy of the information were reviewed by a research assistant assigned by the Research Division of the Instituto Nacional de Cancerología (INC). The study was approved by the INC Ethics Committee under the standards outlined in the Declaration of Helsinki (Fortaleza, Brazil, 2013) and the World Conference on Harmonization for Good Clinical Practices (ICH-GCP E6, 1996) [18]. Since this was a retrospective study without additional interventions beyond the breast cancer treatment protocols followed by the UFM-INC, the signing of informed consent was not required.

### 2.2. Statistical Analysis

A descriptive analysis was conducted to characterize the demographic, clinical, and anatomopathological profiles, as well as the treatment received by the study patients. For quantitative variables, the mean and standard deviation were used for those with a normal distribution, or the median and interquartile range for those with a non-normal distribution. Qualitative variables were presented as absolute and relative frequencies.

The primary outcome of interest was time to recurrence (TR), defined as the interval from the date of the first surgical procedure for breast cancer treatment to the date when any recurrence (local, regional, systemic, or mixed) was confirmed. Overall survival (OS) was defined as the time from the initial surgery to the patient’s death (from any cause) [19]. For patients with no follow-up appointments recorded in the past year, telephone follow-up was conducted during January and February 2025 to assess health status and prevent loss to follow-up. Cases that did not experience the event of interest were right-censored.

The frequency of these outcomes was calculated using incidence rates expressed as events per 100 patient-years, along with their respective 95% confidence intervals (CI). The percentages of local, regional, distant, or mixed recurrence were calculated using the total number of patients in the cohort as the denominator. Pathological response was reported based on the available anatomic pathology (AP) report. In cases where the residual cancer burden (RCB) score was not explicitly reported, it was calculated by inputting the reported parameters into the validated online tool of the MD Anderson Cancer Center protocol for calculating RCB after neoadjuvant treatment [20,21].

The Kaplan–Meier method was used to estimate survival probability, while the Tarone–Ware test was applied for stratified analysis. Additionally, the Restricted Mean Survival Time (RMST) estimator was used to assess the average cumulative survival up to a clinically relevant time point. Cox proportional hazards models were used to calculate hazard ratios (HR) and 95% CIs for each prognostic factor using both univariate and multivariate analyses. Factors were chosen based on clinical expertise and literature review.

Finally, to identify potential factors associated with SLN positivity, a bivariate analysis was conducted on the subset of patients who underwent Upfront SLNB, using the Chi-square test or the Fisher–Freeman–Halton test, as appropriate, for qualitative variables. An unadjusted logistic regression model was used to assess the impact of each predictive variable of interest on the odds ratio (OR), as an indicator of the effect of SLN positivity. All comparisons were two-tailed, and a *p*-value < 0.05 was considered statistically significant. Statistical analyses were conducted using R-Project software^®^, version 4.3.3 (R Foundation for Statistical Computing, Vienna, Austria).

## 3. Results

Between 1 September 2013, and 31 August 2023, a total of 1108 clinical history records of patients who underwent SLNB at the UFM-INC were reviewed. After reviewing these records, 793 patients met the study inclusion criteria. Six patients were excluded from this group: five due to a lack of SLN identification, and one from the Post-NACT SLNB group because she was transferred to another institution after completing systemic treatment. A total of 787 patients were ultimately included: 81.7% (*n* = 643) underwent Upfront SLNB, while 18.3% (*n* = 144) underwent Post-NACT SLNB. Figure 1 details the reasons for exclusion from the final analysis.

### 3.1. Clinicopathological Characteristics

Among the 787 patients included in the study, the average age was 59.9 ± 11.5 years; 80.8% (*n* = 636) were 50 years or older. 62.9% (*n* = 495) were in clinical stage IIA, while 32.4% (*n* = 255) were in stage I. Of the Post-NACT SLNB group, 15 patients (10.41%) were staged as cN1. Regarding tumor biology, the most common biological subtype was luminal A in 50.8% (*n* = 400) of the cases, followed by Luminal B HER2-negative in 31.6% (*n* = 249). According to histopathology, most cases were infiltrating ductal tumors not otherwise specified (NOS) in 87.9% (*n* = 692) and histological grade 2 in 68.6% (*n* = 540). A total of 65.4% (*n* = 515) of patients had T2 tumors. Lymphovascular invasion (LVI) was present in 23.9% (*n* = 188) of cases.

### 3.2. Sentinel Lymph Node Outcome

The SLN identification rate was 99.36% (782/787) across the entire cohort: 99.22% (643/648) for the Upfront SLNB group and 100% for the Post-NACT SLNB group. The five patients in whom the SLN was not identified were excluded from the study. The most commonly used technique for SLN identification was technetium 99 (99mTc) (90.72%; *n* = 714). In most cases (44.4%; *n* = 349), only one SLN was identified, with 28.6% (*n* = 225) testing positive for metastasis. The rate was 32% (*n* = 206) in the Upfront SLNB group and 13.1% (*n* = 19) in the Post-NACT SLNB group.

SLN involvement varied significantly based on tumor size, with positive SLN found in 22% (*n* = 19) of patients with T1a-T1b tumors; 24.4% (*n* = 43) of those with T1c tumors; 31.2% (*n* = 161) of patients with T2 tumors; and 20% (*n* = 2) of patients with T3 tumors.

Among the 225 patients in the cohort with SLN positive for metastasis, macrometastasis involvement (pN1) was reported in 81.3% (*n* = 183) of cases, micrometastasis in 17.3% (*n* = 39) (pN1mi), and only 1.33% (*n* = 3) had isolated tumor cells (pN0(i+)). ALND was omitted in 56% (*n* = 126) of the SLN-positive patients in the Upfront SLNB group. Of the total SLN-positive patients, 99 (43.5%) had an indication for ALND, but it was performed only in 98 because one patient refused the procedure. Of the 98 patients who underwent ALND, 80.6% (*n* = 79) were in the Upfront SLNB group; in this group, 54.4% (*n* = 43) had positive lymph nodes on the ALND AP report, whereas among the 19 (19.4%) patients in the Post-NACT SLNB group, the AP report showed positive nodes in 26.3% (*n* = 5).

Of the 19 patients with positive SLN in the Post-NACT SLNB group, three were staged as cN1, and their tumor biology was luminal B HER2-negative; two of them had RCB II, while the third had RCB III. Similarly, of the 15 patients with cN1, five achieved pathological complete response (pCR), none showed disease progression, and by the end of data collection, all 15 were alive and disease-free. None of the 15 patients with cN1 had clip marking before surgery, due to administrative problems. Among the 16 patients with positive SLN whose lymph node staging was cN0, five were found to have metastatic lymph node involvement on ALND. Therefore, ALND was avoided in 86.81% (*n* = 125) of the patients who received NACT. All patients who had positive SLN underwent ALND.

When analyzing the number of lymph nodes involved during ALND, according to the histopathological report, it was found that in the Upfront SLNB group (*n* = 79), 92.4% (*n* = 73) had macrometastasis. Of these, 34.2% (*n* = 25) had between 1 and 3 positive nodes (pN1), and 19.2% (*n* = 14) had involvement of 4 or more nodes (>pN2). Conversely, the AP report identified micrometastasis (pN1mi) in the remaining 7.6% (*n* = 6) patients: two of them had between 1 and 3 positive nodes (pN1), another two had 4 or more positive nodes (>pN2), and the remaining two showed no tumor involvement in the resected nodes (pN0). No patient had isolated tumor cells in the axillary dissection (pN0(i+)).

In the Post-NACT SLNB group (*n* = 19), 94.7% (*n* = 18) had macrometastasis (ypN+). Of these, 27.8% (*n* = 5) had between 1 and 3 involved nodes (ypN1), while the remaining 13 patients showed no lymph node involvement on ALND (ypN0). The remaining patient presented isolated tumor cells (pN0(i+)) without lymph node involvement on ALND (ypN0). No statistically significant association was found between the type of lymph node involvement in the SLN (macrometastasis or isolated tumor cells) and the presence of positive lymph nodes on ALND (*p* = 0.4357). Regarding extracapsular extension, 35 (15.55%) patients had extracapsular extension (ECE) > 2 mm, of these, 13 had negative nodes on ALND, 12 patients had between 1 and 3 lymph nodes, 8 had between 4 and 9 and 2 had more than 10 positive lymph nodes. On the other hand, 34 (15.11%) patients had ECE < 2 mm, 14 of them underwent ALND omission and the other 20 patients had positive lymph nodes in the ALND. 6 patients with ECE developed distance recurrence, none of them had regional recurrence. Table 1 shows the clinical and histopathological characteristics of the patients included in the study.

Regarding postoperative complications, these were reported in 7.24% (*n* = 57) of the total patients in the entire cohort, with a higher occurrence in the Upfront SLNB group (84.8%, *n* = 48). The most common complication was seroma in 35 patients, followed by superficial surgical site infection in 7 patients, and lymphedema and neuropathic pain in one patient; but due to the retrospective nature of the study, gathering detailed information about lymphedema is difficult. In general, it was found that, by axillary approach, the proportion was 0.071 (95% CI [0.03, 0.14]) for SLNB + ALND and 0.072 (95% CI [0.05, 0.09]) for SLNB, respectively.

### 3.3. Treatment Types

Conservative surgery was performed in 74.2% (*n* = 584) of cases. Additional surgical treatment was needed in 16.5% (*n* = 130) of patients. The most common neoadjuvant chemotherapy (NACT) regimen was a combination of anthracyclines and taxanes (AC-T) in 52.8% (*n* = 76) of patients. A pathological complete response (pCR), measured as RCB 0 (zero) according to the MD Anderson Cancer Center score [20], was observed in 30.56% (*n* = 44) of cases, with a significant difference in HER2 and triple-negative (TNBC) tumors (40% and 35%, respectively). The pathological response rate could not be determined in nine patients because this score was not standardized by the INC pathology group in 2013. 38.5% (*n* = 303) received adjuvant chemotherapy. 21 patients received adjuvant treatment for residual disease (3 T-DM1 and 18 Capecitabine) and 86.7% (*n* = 682) received adjuvant hormone therapy. Adjuvant RT was given to 76.8% (*n* = 605) of patients, among these patients 577 (95.37%) after conservative surgery. Table 2 details the treatment types administered.

Predictive factors for SLN positivity: Age, histological grade, and molecular subtype were not linked to SLN positivity in the subset of patients who underwent Upfront SLNB. In unadjusted logistic regression analysis, locally advanced clinical stages IIB–IIIA (*p* = 0.03), lobular histology (*p* = 0.03), and lymphovascular invasion (LVI) (*p* < 0.01) were associated with positive SLN (Table 3). In the adjusted logistic regression analysis, clinical stages IIB-IIIA (OR 3.01; 95% CI: 1.03–9.07; *p* = 0.04), lobular histology (OR 3.28; 95% CI: 1.09–10.60; *p* = 0.04), and LVI (OR 5.41; 95% CI: 3.64–8.11; *p* < 0.01) were significantly associated with SLN positivity (Figure 2). No multicollinearity problems were found in the variables included in the multivariate logistic regression model for predicting SLN positivity (Variance Inflation Factor -VIF < 1.5).

For the Post-NACT SLNB group, an attempt was made to adjust this analysis using the same variables as the Upfront SLNB group, but due to the small number of patients with positive SLN, the estimates were very unstable. Only lymphovascular invasion was identified as an independent risk factor in this group (OR 15.3; 95% CI: 4.85–52.1).

This study observed an increase in the number of post-NACT SLNB procedures performed. Between 2013 and 2020, 72 post-NACT SLNB procedures were carried out, averaging 9 per year (Appendix A). The same total of 72 procedures was achieved in just three years (2021–2023), with an average of 24 per year, representing a 2.7-fold rise in annual procedures. This shift indicates a major change in clinical practice driven by new guidelines and the growing acceptance of the Post-NACT SLNB approach.

### 3.4. Recurrence Location and Treatment

A total of 60 patients (7.62%) with tumor recurrence were identified in the entire cohort, with most of these patients undergoing initial surgical management (81.6%; *n* = 49). Distant recurrence was the most common, accounting for 58.1% (*n* = 43) of cases: 34 occurred in the Upfront SLNB group and 9 in Post-NACT SLNB patients. Bone was the main site in 24 patients, but 16 of these were mixed (distant lymph node, lung, pleura, or liver). Of the 43 patients with distant recurrence, the SLN result was positive in 30.2% (*n* = 13) of the Upfront SLNB group and 4.6% (*n* = 2) in the Post-NACT SLNB group. Most patients with distant recurrence had luminal B HER2-negative tumors (53.4%; *n* = 23). Consequently, the most frequently used treatment was the combination of iCDK4/6 (Inhibitor of cyclin-dependent kinase 4 and cyclin-dependent kinase 6) plus endocrine therapy (ET) in 34.8% (*n* = 15); 27.9% (*n* = 12) received chemotherapy, and 13.9% (*n* = 6) received endocrine therapy alone.

Regional recurrence occurred in 17 patients (2.16% of the entire cohort). In the Upfront SLNB group, this happened in 15 patients (2.33%). Of these, 11 patients (2.52%) had a negative SLN pathology report, and the other four had a positive SLN (three with omission of ALND and one with ALND). In the Post-NACT SLNB group, only two patients (1.39%) had regional recurrence; in both, the pathology report showed a negative SLN. No recurrence was recorded in the Post-NACT SLNB subgroup with positive nodes who underwent ALND.

Regarding specific axillary recurrence, 16 cases were identified in total, corresponding to 2.03% of the cohort (16/787). Of these, 15 occurred in the Upfront SLNB group: 11 patients had negative SLN and 4 had positive SLN (three with omission of ALND and one with ALND). In the Post-NACT SLNB group, only one patient had axillary recurrence (negative SLN).

Of the 17 patients with regional disease progression, 9 experienced only regional recurrence, while 8 experienced regional recurrence along with another site, which is referred to as mixed recurrence, including both regional and distant recurrence. In the first case (only regional recurrence), when characterized by specific SLN behavior, it was found that 55.6% (*n* = 5) were patients with Upfront SLNB (negative SLN), 22.2% (*n* = 2) had Upfront SLNB + omission of ALND (positive SLN), 11.1% (*n* = 1) had Upfront SLNB + ALND (positive SLN), and 11.1% (*n* = 1) had Post-NACT SLNB (negative SLN). None of these patients with only a regional recurrence underwent Post-NACT SLNB + ALND (positive SLN).

In the second case (regional recurrence with another site), 75.0% (*n* = 6) of the patients underwent Upfront SLNB (SLN negative), 12.5% (*n* = 1) had Upfront SLNB + omission of ALND (SLN positive), and 12.5% (*n* = 1) underwent Post-NACT SLNB (SLN negative). As in the first case, none of these patients underwent Post-NACT SLNB + ALND (SLN positive). The predominant tumor biology in this group was luminal B HER2-negative (72.7%; *n* = 8). Of the 12 patients with axillary recurrence, 9 underwent ALND, and the other 3 received RT for axillary and supraclavicular involvement. On the other hand, in the two Post-NACT SLNB patients with regional recurrence, one was axillary and the other infraclavicular. The patient with axillary recurrence had stage IIIA triple-negative breast cancer; the SLNB had been negative, and she underwent ALND plus systemic chemotherapy. Only three patients (2.38%; *n* = 3/126) with omission of ALND presented axillary recurrence; all of them received adjuvant chemotherapy and had luminal B HER2-negative tumors. One of them died 9 years after diagnosis (mixed recurrence), and the other two are alive with controlled disease. Table 4 shows the type, location, and treatment of recurrences.

### 3.5. Time to Recurrence (TR) and Overall Survival (OS)

The median follow-up time was 51.8 months (IQR: 21.0–84.5). During the follow-up period, 7.62% (*n* = 60) of patients experienced some type of recurrence; 8.51% (*n* = 67) had died: 39 (58.2%) due to the disease and 28 (41.8%) died from another cause. Additionally, 3.60% (*n* = 28) of all patients included in the study were alive with disease; 1.40% (*n* = 11) were determined to be lost to follow-up, and 86.5% (*n* = 681) were alive without disease. At 60 months, the probability of being recurrence-free for the entire cohort was 91.4% (95% CI: 88.9–93.9), and the overall survival (OS) was 96.1% (95% CI: 94.5–97.7). The median follow-up time for patients with luminal molecular subtypes undergoing Post-NACT SLNB was 30.5 months (IQR: 15.8–55.1), while for patients with Upfront SLNB it was 57.0 months (IQR: 22.9–88.6).

The median OS for the entire cohort was not estimable in either oncological outcome. Kaplan–Meier analysis by subgroups (Upfront SLNB and Post-NACT SLNB) did not reveal significant differences in the treatments considered for SLN management in TR and in OS. The probability of being free of recurrence was 93.0% (95% CI: 90.0–96.1) in patients with Upfront SLNB (negative SLN), 90% (95% CI: 83.4–97.2) in Upfront SLNB + Omission of ALND (positive SLN), and 92.8% (95% CI: 87.0–99.1) in Upfront SLNB + ALND (positive SLN). The OS rates were 91.8% (95% CI: 88.5–95.1) in Upfront SLNB (negative SLN), 96.2% (95% CI: 91.9–100) in Upfront SLNB + Omission of ALND (positive SLN), and 94.0% (95% CI: 88.4–99.9 in Upfront SLNB + ALND (positive SLN) (*p* > 0.05) (Figure 3, panels B–E).

The probability of being recurrence-free was 84.8% (95% CI: 74.1–97.2) in patients with Post-NACT SLNB and 71.4% (95% CI: 44.7–100) in Post-NACT + ALND (SLN positive), while the OS rate was 91.5% (95% CI: 84.9–98.6) in Post-NACT SLNB (SLN negative) patients and 100% (95% CI not estimable) in Post-NACT SLNB + ALND (SLN positive), respectively (*p* > 0.05) (Figure 3, panels C–F), since none of these patients had died. Only 42.1% had a 60-month follow-up, a follow-up time that is insufficient, especially for luminal tumors (HR+), which are the majority of the patients in the cohort. In this case, the difference in RMST for the time-to-death outcome was 5.47 months (95% CI: 1.79–9.14). The point estimate indicated that, on average, patients with luminal subtypes survived 5.47 months longer than those in the non-luminal group after a minimum follow-up of 60 months, with this difference being statistically significant (*p* < 0.01). On the other hand, the difference in RMST for time to relapse was 2.77 months (95% CI: 0.74–6.28), indicating that, on average, patients with luminal subtypes survived 2.77 months longer than the non-luminal group after at least 60 months of follow-up. However, this difference was not statistically significant (*p* = 0.123), suggesting that longer follow-up is necessary for patients with luminal molecular subtypes.

Regarding SLN positivity (SLN+ vs. SLN-) (Figure 4), no statistically significant differences were observed in prognosis in the two outcomes evaluated. However, a significant impact was evident when considering the number of involved lymph nodes in the TR outcome, where patients with four or more positive lymph nodes had a higher risk of recurrence compared to patients with negative SLN (HR 2.70; 95% CI: 1.12–6.49, *p* = 0.04). It should be noted that the 16 patients with more than four positive lymph nodes were in the Upfront SLNB group, but in the overall cohort, the number of lymph nodes had no impact on this outcome. On the other hand, the biological subtype at the time of diagnosis was a strong determinant in the oncological outcomes (*p* < 0.01) (Figure 3, panels A and D), with the triple-negative subtype being the one with the worst prognosis compared to the luminal A biological subtype in TR (HR 4.73; 95% CI: 2.02–11.1; *p* < 0.01) and in OS (HR 3.87; 95% CI: 1.92–7.79; *p* < 0.01). The incidence density rate of recurrence was 1.88 recurrence events per 100 patient-years (95%CI: 1.43–2.42), while the rate of death was 2.00 per 100 patient-years (95%CI: 1.55–2.54).

In the unadjusted analysis, histological grade and biological subtype were independently associated with a worse prognosis in TR. In OS, histological grade, biological subtype, and type of breast surgery were associated with a worse prognosis (*p* < 0.05) (Table 5).

On the other hand, the adjusted or multivariate analysis, based on selected clinical and pathological characteristics, showed that tumor size (T2 vs. T0) (HR 3.12; 95% CI: 1.01–9.59; *p* = 0.05), histological grade 2 vs. 1 (HR 4.30; 95% CI: 1.01–18.4; *p* = 0.05), and the biological subtypes luminal B HER2-negative (HR 2.90; 95% CI: 1.55–5.44; *p* < 0.01) and triple-negative (HR 5.90; 95% CI: 1.90–18.3; *p* < 0.01), compared to luminal A, were significantly linked to a worse prognosis for breast cancer recurrence (Figure 4, panel A). For overall survival (OS), histological grade (grade 2 vs. grade 1) (HR 3.14; 95% CI: 1.19–8.24; *p* = 0.02), biological subtype (triple-negative vs. luminal A) (HR 3.97; 95% CI: 1.48–10.6; *p* < 0.01), and type of breast surgery (mastectomy vs. breast-conserving surgery) (HR 2.20; 95% CI: 1.30–3.73; *p* < 0.01) were significantly associated with an increased risk of mortality (Figure 4, panel B). No significant associations were identified for lymph node involvement, initial treatment type, or axillary surgery type in either outcome. Both models met the Cox proportional hazards assumption (Appendix A).

## 4. Discussion

This cohort study of 787 women details the experience of de-escalating axillary surgery using the SLNB technique at a leading cancer center in Colombia. Throughout the entire study period (2013–2023), 225 patients with a positive SLN were identified; 170 of these patients were found between 2013 and 2020. Axillary lymph node dissection (ALND) was omitted in 50% (*n* = 85) of cases. The remaining 55 patients were identified between 2021 and 2023, with ALND omitted in 74.5% (*n* = 41) of this group. Meanwhile, the study observed an increase in the number of post-NACT SLNB procedures performed. Between 2013 and 2020, 72 post-NACT SLNB procedures were carried out, averaging 9 per year (Appendix A). The same total of 72 procedures was achieved in just three years (2021–2023), with an average of 24 per year, representing a 2.7-fold rise in annual procedures. This shift indicates a major change in clinical practice driven by new guidelines and the growing acceptance of the Post-NACT SLNB approach.

Since 2021, a gradual shift towards de-escalation strategies for axillary surgery has been observed, in line with recommendations from international guidelines and supported by recent evidence, such as that from the SENOMAC study [12]. Unlike previous trials, such as the ACOSOG Z0011 study [6] and the IBCSG 23-01 trial [9], which mainly included patients undergoing breast-conserving surgery and radiotherapy, the study by Boniface et al. [12] expanded the inclusion criteria to include patients with one or two SLNs that are exclusively positive for macrometastasis (>2 mm), even if they underwent mastectomy (36%), and without excluding extracapsular spread. This greatly improved the relevance of its findings to clinical practice. With over 2500 patients and a median follow-up of 46.8 months, the SENOMAC study, similar to the ACOSOG Z0011 and IBCSG 23-01 trials, showed that omitting ALND does not compromise survival or increase locoregional recurrence rates, and is also linked to a significant reduction in surgical morbidity [6,9,12].

In this cohort, an overall SLN identification rate of 99.3% was achieved, with only five patients excluded due to failure to detect the SLN intraoperatively. This figure is comparable to those reported in landmark studies, such as the NSABP B-32 trial (97.2%) [5], the AMAROS study (97%) [10], the ALMANAC study (96.9%) [4], Veronesi et al. (96%) [21], and the Upfront SLNB arm of the SENTINA study (99.1%) [15]. The identification rate was 99.22% (643/648) in the Upfront SLNB group, reaching 100% in the Post-NACT SLNB group. This result was facilitated by the systematic use of a dual-label technique (dye and radiotracer) and careful patient selection. This rate is comparable to the 91.2% reported in the AXSANA study for Post-NACT cN1 patients, although it should be noted that in that study, the localization rate varied depending on the technique used, reaching up to 96% with optimal marking methods [22].

In this study, the overall SLN positivity rate was 28.6% (*n* = 225), higher in the group undergoing Upfront SLNB (32%, *n* = 206), which is slightly above the rates reported in reference studies such as the NSABP B-32 trial (26%) [5] and the ALMANAC study (25%) [4]. In most procedures (44.4%, *n* = 349), only one SLN was identified, which could limit the detection of nodal disease and influence the observed positivity rate. This difference may also relate to a higher proportion of T2–T3 stage tumors (68.4%) compared to approximately 30% reported in the studies used for comparison, as well as the presence of biological subtypes associated with worse prognosis, such as HER2-positive (9.6%) and triple-negative (7.5%) tumors, which have been linked to a higher nodal burden in the literature [23]. However, in this analysis, neither biological subtype, age, nor histological grade showed a statistically significant association with nodal positivity in the group of patients undergoing Upfront SLNB.

Sentinel lymph node involvement was significantly linked to lymphovascular invasion (LVI) (OR 5.41; 95% CI: 3.64–8.11; *p* = 0.04), invasive lobular carcinoma histology (OR 3.28; 95% CI: 1.09–10.6; *p* = 0.04) and clinical tumors ≥ 2 cm (stages IIB–IIIA) (OR 3.01; 95% CI: 1.03–9.07; *p* = 0.04). These results align with those of Viale et al. [24], who identified peritumoral vascular invasion as the main predictor of SLN metastasis (OR 5.26; 95% CI: 4.44–6.23; *p* < 0.0001), followed by tumor size, multifocality, and unfavorable histology [24]. A meta-analysis of 28 studies also showed a higher frequency of LVI in patients with positive lymph nodes (45.85% vs. 23.85%), with a significant correlation (*r* = 0.24) [25].

Although lobular histology showed an association with positive SLN status in the entire cohort, its value as an independent predictor remains controversial. Its biological features tend to promote multifocality and multicentricity, which could account for a higher lymph node burden. Some studies consider it an independent predictor [26,27], but others find that this association vanishes after adjusting factors such as tumor size and multifocality [28]. Nonetheless, LVI, lobular histology, and tumor size are included in validated predictive models, like the nomograms from the Memorial Sloan-Kettering Cancer Center (MSKCC) [29] and the MD Anderson Cancer Center [30], which are helpful for estimating lymph node risk, particularly in resource-limited settings.

In the Post-NACT SLNB group, SLN positivity was 13.1% (*n* = 19), lower than in studies focusing on patients who converted from cN1 to cN0, such as ACOSOG Z1071 (31%) [13] and SENTINA (24%) [15], but comparable to the GANEA 2 study (17.7%) [16], which mainly included cN0 patients (78%), a profile similar to this cohort.

The estimated average number of SLNs identified per patient was 1.94, with a single lymph node detected in 44.4% of all cases in the study. These figures are similar to those reported in reference studies such as the ACOSOG Z0011 [6] study, which found an average of 2 lymph nodes, the SENTINA study [15], which reported a median of 2 lymph nodes, and the SN FNAC study [14], with an average of 2.1 lymph nodes. The percentage of patients with only one identified lymph node (44.4%) was slightly higher than that reported in the ACOSOG Z0011 study [6] (36%) and SN FNAC (41%) [14].

The analysis of axillary residual burden in patients with positive SLNB who underwent ALND showed nodal involvement in 54.4% of patients in the Upfront SLNB group and in 26.3% of the Post-NACT SLNB group, demonstrating a 52% relative reduction in the latter group, which reflects the effect of systemic therapy. This lower axillary burden (RCB-0) was associated with a pCR rate of 30.6% and a low incidence of extensive residual disease (RCB-3: 7.6%).

In our cohort, 54.4% of patients with a positive SLN who underwent ALND had additional involvement of non-sentinel lymph nodes. This rate exceeds that reported in relevant clinical trials, such as the ACOSOG Z0011 study [6], where only 27.3% of patients with one or two positive SLNs undergoing ALND had additional metastases; the AMAROS study [10], with a rate of 33%, and the IBCSG 23-01 trial [9], which included only patients with micrometastasis and reported 13%. In the SENOMAC study [12], the rate of additional nodal involvement was 34% when there was a single positive SLN and increased to 51.3% with two positive SLNs. The high rate of residual lymph node disease in our series may relate to the clinical characteristics of the cohort, which included a high proportion of T2–T3 tumors (67%), LVI in 23.9% (associated with positive SLNB) (OR 5.33; 95% CI: 3.63–7.88), lobular histology in 5.5% (OR 3.83; 95% CI: 1.38–11.7), and identification of only a single SLN in 44.3% of cases, which could reflect a clinical selection bias toward patients with a higher suspected axillary burden.

Predictive models, such as the MSKCC nomogram (2003) [29], estimate a 50 to 70% risk when there are more than two positive SLNs or tumors larger than 2 cm. The 26.3% recurrence rate observed after Post-NACT SLNB is similar to rates reported in studies such as FNAC (21.8%) [14], GANEA 2 (24.2%) [16], SENTINA (20.9–33.8%) [15], and ACOSOG Z1071 (39%) [13], all of which are below the 40–60% range described for Upfront SLNB.

In patients with complete nodal response (ypN0) after NACT, ALND was omitted in 86.8% of cases. Montagna et al. (2024) [31] reported an omission rate of 99.7% in cN+ patients who converted to ypNO status through SLNB and/or targeted axillary dissection (TAD), with only 1% of axillary recurrence at 5 years. The GANEA 2 trial [16] showed an omission rate of 74%, and the SN-FNAC study [14] reported 65.6%, further supporting the safety of omitting ALND in this setting.

There were no statistically significant differences in recurrence rates between the groups undergoing ALND and those treated with SLNB alone (negative SLNB result) or with omission of ALND (positive SLNB result). At 60 months, the recurrence-free survival (RFS) rate was 91.4% (95% CI: 88.9–93.9%), with no differences between the different surgical strategies assessed. This outcome aligns with the findings of the SENOMAC study [12], which reported similar RFS rates between SLNB and ALND (92.9% vs. 92.0%). In this cohort, recurrence was mainly associated with intrinsic tumor characteristics, such as triple-negative subtype, T2 tumors, high histological grade, and luminal B HER2-negative subtype.

The time to recurrence (TR) was favorable, with a cumulative probability of 91.4% (95% CI: 88.9–93.9%) of being recurrence-free at 60 months. This finding aligns with data from key studies: the NSABP B-32 trial [5] (mean follow-up of 95.6 months; regional recurrence < 1%), Veronesi et al. (~102 months; DFS: ~89–90%) [21], the ACOSOG Z0011 study (median: 112 months; DFS: ~80–83% at 10 years) [6], and the SENOMAC study (66 months; axillary recurrence 1.0% without SLNB vs. 0.8% with SLNB) [12]. Furthermore, in low-risk settings, studies on omitting SLNB, such as SOUND (DFS: 97.7% at 60 months) [32] and INSEMA (DFS: 91.9% without SLNB vs. 91.7% with SLNB), show similar results [33]. Taken together, these data are consistent with current guidelines, such as ASCO (2025), which already recommend considering the omission of SLNB in patients with up to T2 disease and a clinically negative axilla, provided proper patient selection [34].

In the multivariate analysis, several independent predictors of recurrence were identified. T2 tumors had a 3.12-fold higher risk of recurrence compared to T1 tumors (HR 3.12; 95% CI: 1.01–9.59; *p* = 0.047). Likewise, tumors with histological grade 2 showed a 4.3-fold higher risk of recurrence compared to those with grade 1 (HR 4.30; 95% CI: 1.01–18.4; *p* = 0.049). Concerning biological subtypes, luminal B HER2-negative tumors were associated with nearly three times the risk of recurrence compared to luminal A tumors (HR 2.90; 95% CI: 1.55–5.44; *p* < 0.01), while the triple-negative subtype had the worst prognosis, with almost six times the risk (HR 5.90; 95% CI: 1.90–18.3; *p* < 0.01).

The OS rate at 60 months in this cohort was 96.1% (95% CI: 94.5–97.7), consistent with prospective studies like SENOMAC [12] (93% with SLNB vs. 92% with ALND), SINODAR-ONE [35] (98.8% vs. 98.9%), and ACOSOG Z0011 [6] (86.3% with SLNB vs. 83.6% with ALND), which supports the oncological safety of omitting ALND in selected patients, with no negative effect on long-term survival.

In the multivariate analysis, histological grade 2 was identified as an independent predictor of lower survival, with a 3.14-fold increased risk compared to grade 1 (HR 3.14; 95% CI: 1.19–8.24; *p* = 0.02), as was the triple-negative subtype, with a risk nearly four times higher than the luminal A subtype (HR 3.97; 95% CI: 1.48–10.6; *p* < 0.01). Similarly, the type of breast surgery was significantly associated with OS: patients who underwent mastectomy had a higher risk of death compared to those treated with breast-conserving surgery (HR 2.20; 95% CI: 1.30–3.73; *p* < 0.01), likely reflecting a greater initial tumor burden or more aggressive biological characteristics.

These findings reinforce the idea that tumor biology and histopathological variables are the main determinants of oncological prognosis and disease progression, even in cases with a low axillary lymph node burden. The association between mastectomy and decreased survival can be viewed as an indirect sign of more advanced disease, considering its common use in large, multifocal tumors or in patients ineligible for radiotherapy. Conversely, factors such as age, type of axillary surgery, lymph node status, and initial treatment were not significantly associated with OS, unlike in studies such as the ACOSOG Z0011 study [6], where the presence of multiple positive lymph nodes was a key predictor of mortality.

This study did not find statistically significant differences in oncological outcomes based on the type of axillary management. In the Upfront SLNB group, the 60-month recurrence rates were similar among patients with negative lymph nodes (93.0%), positive lymph nodes without ALND (90.0%), and those with ALND (92.8%) (*p* > 0.05), as were the OS rates (91.8%, 96.2%, and 94.0%, respectively; *p* > 0.05). In the Post-NACT SLNB group, this pattern continued: the recurrence rate was 84.8% in ypSLN- and 71.4% in ypSLN+ with ALND; OS was 91.5% vs. 100% (both *p* > 0.05). Multivariate analysis confirmed that the extent of axillary surgery was not an independent predictor for TR (HR 1.22; 95% CI: 0.63–2.35; *p* = 0.55) or OS (HR 0.88; 95% CI: 0.45–1.73; *p* = 0.71). These findings support that omitting ALND in cases of negative SLNB results does not compromise oncological outcomes, which primarily depend on tumor biology—such as molecular subtype and histological grade—as the main prognostic factors.

In this cohort, the overall regional recurrence rate was 2.16%, with an axillary-only recurrence rate of 1.87%. Importantly, these recurrence rates fall within the range reported in pivotal clinical trials supporting axillary surgical de-escalation, thereby reinforcing the safety and applicability of this approach—even in middle-income settings. In the subgroup analysis, patients who underwent Upfront SLNB had a regional recurrence rate of 2.33% and an axillary recurrence rate of 1.85%, while in the Post-NACT SLNB group, the rates were 1.39% and 2.00%, respectively. These figures fall within expected ranges and align with studies supporting surgical de-escalation. In the ACOSOG Z0011 study [6], axillary recurrence was 0.9%, and regional recurrence remained below 3% at 10 years, even without ALND in patients with 1 to 2 positive lymph nodes. Similarly, the IBCSG 23-01 trial [9], which included patients with micrometastasis, reported a regional recurrence rate of 1.3%. The AMAROS trial [10] showed that axillary radiotherapy offers locoregional control comparable to ALND, with 10-year axillary recurrence rates of 0.93% in the ALND group and 1.82% in the radiotherapy group. In the Post-NACT setting, the Z1071 study [13] reported regional recurrences below 1.5% in patients undergoing ALND following SLNB, findings confirmed by other recent series.

Several studies have supported the safety of omitting ALND in patients with negative SLN status (pN0). In the NSABP B-32 trial [5], the regional recurrence rate was less than 1%. Similarly, the ALMANAC study [4] reported an axillary recurrence rate of 0.9% with SLNB as the sole procedure. Veronesi et al. [21] found an axillary recurrence rate of 1.85% without axillary dissection, compared to 0.37% with dissection, at 10-year follow-up. Similar results have been reported in studies such as SENTINA [15] and GANEA 2 [16], with axillary recurrence rates of 0.4% and ~0.2%, respectively, consistent with the findings of this cohort.

These figures align with those reported in numerous international multicenter studies and meta-analyses, reinforcing the external validity of our results and the oncological safety of omitting ALND in selected patients with negative SLN status, both in the context of primary (Upfront) SLNB and after neoadjuvant chemotherapy (Post-NACT SLNB).

This study contributes to the growing body of evidence supporting de-escalation of axillary surgery as a safe approach, aligned with global efforts to reduce surgical morbidity without compromising locoregional control. While ALND remains the standard of care in cases with residual disease after Post-NACT SLNB [36], observational studies like this one provide relevant clinical data while awaiting the results of ongoing randomized trials, such as Alliance A011202 [37], ADARNAT [38], and TAXIS [39], which will better define the safety limits for omitting ALND in patients with residual axillary lymph node disease (ypN+).

### Strengths and Limitations

The main strengths of this study include the large number of patients, the refined and standardized surgical technique, as evidenced by the SLN identification rates, and the observed oncological outcomes. Among its limitations, the retrospective design and non- randomized assignment introduce risk of confounding by indication, besides it restricts control over confounding variables, clinical stage and biology differed between groups, limiting direct comparisons. The follow-up period may be too short to detect late recurrences, especially in luminal subtype tumors. The absence of data on morbidity (lymphedema and neuropathic pain) prevents a complete assessment of the functional benefits of surgical de-escalation. Additionally, the small sample size in subgroups like Post-NACT SLNB + ALND and patients with cN1 reduce the statistical power to draw firm conclusions in this clinical context. Despite these limitations, the study supports the idea that tumor biology, rather than the extent of axillary surgery, determines oncological prognosis.

## 5. Conclusions

Sentinel lymph node biopsy, whether performed during primary surgery (Upfront SLNB) or after neoadjuvant chemotherapy (Post-NACT SLNB), is a potentially safe procedure from an oncological perspective in patients with early-stage and locally advanced breast cancer who have responded well to neoadjuvant systemic treatment. Prospective studies are recommended to provide conclusive evidence. Time to recurrence (TR) and overall survival (OS) are related to tumor-specific clinical factors such as the initial clinical stage, tumor burden, and biological subtype, and not to the extent of axillary surgery.

## Figures and Tables

**Figure 1 cancers-17-03396-f001:**
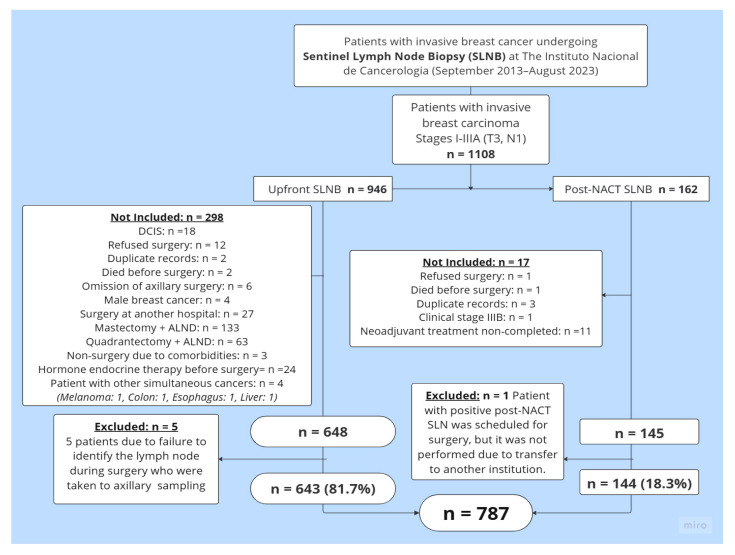
Patient selection flowchart. **Abbreviations****:** SLNB = sentinel lymph node biopsy; DCIS = ductal carcinoma in situ; ALND = axillary lymph node dissection; Upfront SLNB = sentinel lymph node biopsy at the time of the initial surgery, before the administration of any systemic therapy such as neoadjuvant chemotherapy or endocrine therapy; Post-NACT SLNB = sentinel lymph node biopsy after neoadjuvant chemotherapy.

**Figure 2 cancers-17-03396-f002:**
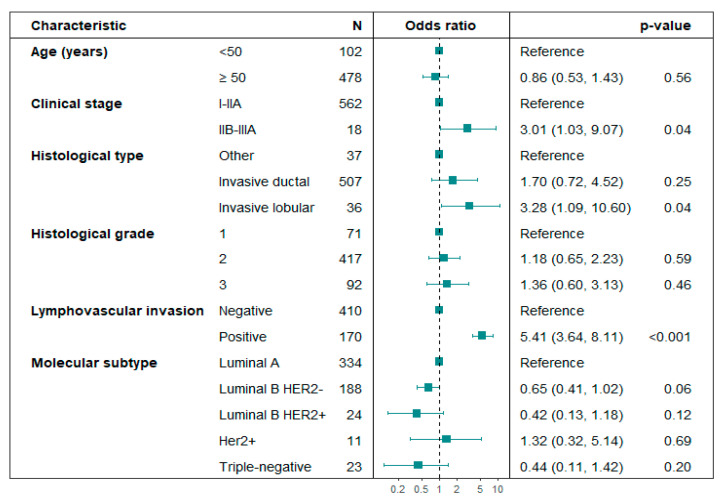
Forest plot. Multivariate analysis of variables associated with axillary lymph node positivity in patients undergoing Upfront SLNB. Hosmer–Lemeshow test *p*-value = 0.984; the likelihood ratio test *p*-value < 0.01.

**Figure 3 cancers-17-03396-f003:**
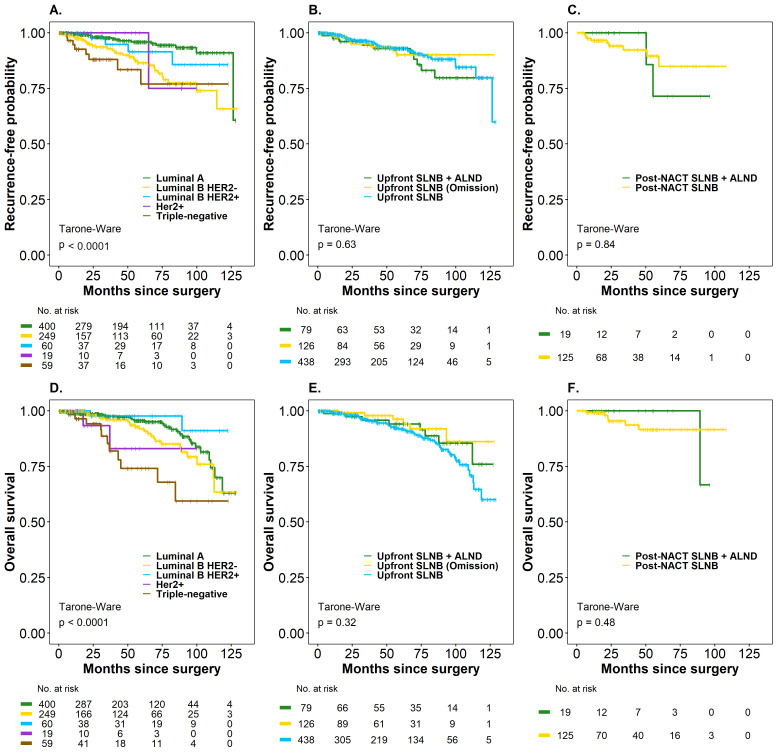
Time to recurrence (TR) and overall survival (OS) (Kaplan–Meier curves) in breast cancer patients according to molecular subtype and treatment type (Upfront SLNB and Post-NACT SLNB), considering axillary management. (**A**) Recurrence-free probability by molecular subtype (Luminal A, Luminal B HER2–, Luminal B HER2+, HER2+, and triple-negative). (**B**) Recurrence-free probability by axillary management in upfront surgery (Upfront SLNB + ALND, Upfront SLNB with omission, and Upfront SLNB). (**C**) Recurrence-free probability by axillary management in post-NACT surgery (SLNB post-NACT + ALND and SLNB post-NACT). (**D**) Overall survival by molecular subtype. (**E**) Overall survival by axillary management in upfront surgery. (**F**) Overall survival by axillary management in post-NACT surgery.

**Figure 4 cancers-17-03396-f004:**
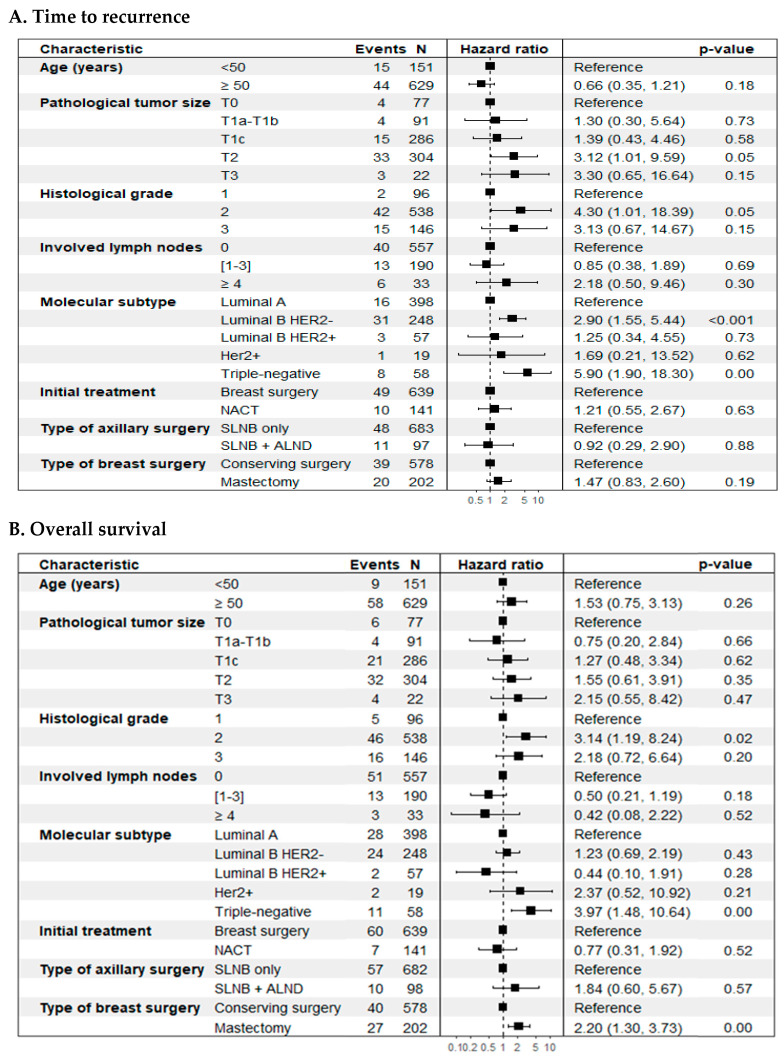
Forest plot. Cox multiple regression analysis was performed to identify significant prognostic factors for time-to-event outcomes. (**A**) Time to recurrence (TR); (**B**) Overall survival (OS).

**Table 1 cancers-17-03396-t001:** Main characteristics of patients diagnosed with breast cancer undergoing SLNB (*n* = 787).

Clinical and Histopathological Data		
**Age (years)**	**Mean ± SD**	59.9 ± 11.5
**Age (years),** ***n*** **(%)**	<50	151 (19.2)
	≥50	636 (80.8)
**Tumor size (T),** ***n*** **(%)**	cT1a-cT1b	86 (10.9)
	cT1c	176 (22.4)
	cT2	515 (65.4)
	cT3	10 (1.30)
**Node involvement (N),** ***n*** **(%)**	N0	772 (98.1)
	N1	15 (1.90)
**Clinical stage,** ***n*** **(%)**	I	255 (32.4)
	IIA	495 (62.9)
	IIB	34 (4.30)
	IIIA	3 (0.40)
**Histological type,** ***n*** **(%)**	Ductal (NOS)	692 (87.9)
	Lobular	43 (5.50)
	Other	52 (6.60)
**Histological grade,** ***n*** **(%)**	1	149 (18.9)
	2	540 (68.6)
	3	92 (11.7)
	No data	6 (0.80)
**Molecular subtype,** ***n*** **(%)**	Luminal A	400 (50.8)
	Luminal B HER2-	249 (31.6)
	Luminal B HER2+	60 (7.60)
	HER2+	19 (2.40)
	Triple-negative	59 (7.50)
**Prognostic data of the surgical specimen**		
**Pathological tumor size,** ***n*** **(%)**	T0	77 (9.80)
	T1a-T1b	91 (11.6)
	T1c	286 (36.3)
	T2	304 (38.6)
	T3	22 (2.80)
	No data	7 (0.90)
**Histological grade ^†^,** ***n*** **(%)**	1	96 (12.2)
	2	543 (69.0)
	3	148 (18.8)
**Lymphovascular invasion,** ***n*** **(%)**	Positive	188 (23.9)
	Negative	529 (67.2)
	No data	70 (8.89)
**Sentinel lymph node technique,** ***n*** **(%)**	99mTc	714 (90.72)
	Methylene Blue	50 (6.40)
	Dual	23 (2.90)
**Number of sentinel lymph nodes identified,** ***n*** **(%)**	1	349 (44.3)
	2	236 (30.0)
	≥ 3	202 (25.7)
**Sentinel lymph node report,** ***n*** **(%)**	Negative	562 (71.4)
	Positive	225 (28.6)
*Sentinel lymph node result ^‡^*	*Macrometastasis*	*183 (81.3)*
	*Micrometastasis*	*39 (17.3)*
	*Isolated tumor cells*	*3 (1.33)*
*Number of positive lymph nodes ^‡^*	*1*	*137 (60.9)*
	[2,3]	*55 (24.4)*
	*≥4*	*33 (14.7)*
**Positive lymph node and extracapsular extension (ECE),** ***n*** **(%)**	*SLNB positive without ECE*	*156 (69)*
	*SLNB positive with ECE < 2 mm*	*34 (15.1)*
	*SLNB positive with ECE > 2 mm*	*35 (15.5)*
**Number of lymph nodes in ALND**		
*Upfront SLNB (n = 79)*	*0*	*36 (45.6)*
	[1–3]	*27 (33.2)*
	*≥4*	*16 (20.2)*
*Post-NACT SLNB (n = 19)*	*0*	*14 (73.7)*
	[1–3]	*5 (26.3)*
	*≥4*	*0 (0.00)*
**Margin involvement,** ***n*** **(%)**	Negative	748 (95.0)
	Positive	39 (5.00)
*Margin status ^¶^*	*1 margin*	*30 (76.9)*
	*2 margins*	*7 (17.9)*
	*3 margins or more*	*2 (5.13)*
**Residual Cancer Burden (RCB) •**		
RCB class	RCB-0	44 (30.6)
	RCB-1	10 (6.94)
	RCB-2	70 (48.6)
	RCB-3	11 (7.64)
	No data	9 (6.25)

**Abbreviations:** SD: standard deviation; SLNB: sentinel lymph node biopsy; ALND: axillary lymph node dissection; NACT: neoadjuvant chemotherapy; RCB: Residual Cancer Burden. †: Nottingham classification; ‡: Percentage over those patients with positive sentinel lymph node report; ¶: Percentage over those patients with positive resection margins; • Percentage over those patients who underwent neoadjuvant chemotherapy (*n* = 144).

**Table 2 cancers-17-03396-t002:** Treatment types administered to patients in the cohort (*n* = 787).

Treatment Type		
**Initial** **treatment, ^†^** ***n*** **(%)**	NACT	144(18.3)
	Breast surgery	643 (81.7)
**Neoadjuvant chemotherapy regimen, ^†^** ***n*** **(%)**		**144 (18.3)**
	*AC-T*	*76 (52.8)*
	*AC-TH*	*22 (15.3)*
	*TRAIN 2*	*11 (7.64)*
	*AC-TC*	*6 (4.17)*
	*Other*	*29 (20.1)*
**Type of breast surgery, ^†^** ***n*** **(%)**		
	Conservative surgery *	584 (74.2)
	Mastectomy	203 (25.8)
**Type of axillary surgery, ^†^** ***n*** **(%)**		
	SLNB only	689 (87.5)
	SLNB + ALND	98 (12.5)
**Management of positive sentinel lymph node, ^†^** ***n*** **(%)**		**225 (28.5)**
*Type* *of management*	*Omission of dissection ***	*126 (56.0)*
	*ALND*	*98 (43.5)*
	*Patient refused ALND*	*1 (0.44)*
**Additional surgical treatments, ^†^** ***n*** **(%)**		**130 (16.5)**
*Treatment type*	*ALND only*	*90 (69.2)*
	*Margin expansion only*	*25 (19.2)*
	*ALND + margin expansion*	*6 (4.62)*
	*Simple mastectomy*	*5 (3.85)*
	*Simple mastectomy + margin expansion*	*2 (1.54)*
	*ALND + simple mastectomy*	*2 (0.76)*
**Adjuvant systemic therapy agents, ^†^** ***n*** **(%)**		* **303 (38.5)** *
	*AC*	*31 (10.2)*
	*AC-T*	*113 (37.2)*
	*AC-TH*	*17 (5.6)*
	*Capecitabine*	*18 (5.9)*
	*TC*	*55 (18.1)*
	*T-DM1*	*3 (0.9)*
	*Trastuzumab monotherapy*	*33 (10.8)*
	*Other chemotherapy regimens*	*33 (10.8)*
**Adjuvant endocrine therapy, ^†^** ***n*** **(%)**		**682 (86.7)**
	*Tamoxifen*	*269 (39.4)*
	*AI*	*238 (34.9)*
	*Tamoxifen + AI*	*164 (24.0)*
	*Other*	*11 (1.61)*
**Adjuvant radiotherapy, ^†^** ***n*** **(%)**		**605 (76.8)**
*Used dose*	*5 fractions 5 Gy ea.*	*70 (11.6)*
	*16 fractions 2.6 Gy ea.*	*303 (50.1)*
	*25 fractions 2 Gy ea.*	*143 (23.6)*
	*Other*	*86 (14.2)*
	*No data*	*3 (0.49)*
**Details on radiotherapy administered to patients with positive SLN and omission of ALND, ^†^** ***n*** **(%)**	*patients with positive SLN and omission of ALND*	* **126 (16)** *
*Type of radiotherapy administered*	*Standard fractionation regimen* *(2 Gy×25 fractions = 50 Gy)*	*55 (43.6)*
	*Hypofractionated regimen* *(2.66 Gy×16 fractions = 42.5 Gy)*	*69 (54.7)*
	*Ultra-hypofractionated regimen* *(5.5 Gy × 5 fractions = 26 Gy)*	*2 (1.5)*
*Radiotherapy technique*	*IMRT* *3D-CRT* *IGRT* *VMAT* *IORT*	*90 (73)* *30 (23.8)* *2 (1.5)* *1 (0.7)* *1 (0.7)*
*Irradiated fields*	*Breast—Axilla*	*10 (7.9)*
	*Breast—Axilla—Supraclavicular fossa*	*57 (45.2)*
	*Breast—Axilla—Supraclavicular fossa—Internal mammary chain*	*5 (3.9)*
	*Chest wall—Axilla*	*1 (0.7)*
	*Chest wall—Axilla—Supraclavicular fossa*	*44 (34.9)*
*High tangential fields*		*9 (7.1)*

**Abbreviations:** 3D-CRT: Three-Dimensional Conformal Radiation Therapy; AC: anthracycline, cyclophosphamide; AC-T: Doxorubicin + Cyclophosphamide followed by Paclitaxel or Docetaxel; AC-TH: Doxorubicin + Cyclophosphamide followed by Paclitaxel/Docetaxel + Trastuzumab; AI: aromatase inhibitor; ALND: axillary lymph node dissection; IGRT: Image-Guided Radiation Therapy; IMRT: Intensity-Modulated Radiation Therapy; IORT: Intraoperative Radiotherapy; NACT: neoadjuvant chemotherapy; SLNB: sentinel lymph node biopsy; T: taxanes; T-DM1: Trastuzumab emtansine; TC: taxane and cyclophosphamide; TH: taxanes and trastuzumab; VMAT: Volumetric Modulated Arc Therapy. †: Percentage over the entire cohort of patients (n = 787); *: Includes quadrantectomy and oncological mammoplasty; **: Only in Upfront SLNB.

**Table 3 cancers-17-03396-t003:** Factors predictive of positive lymph nodes (SLNB)—unadjusted analysis.

	Lymph Nodes			
Characteristic	Positive	Negative	OR [95% CI]	*p*-Value
**Age (years),** ***n*** **(%)**				
<50	41 (19.9)	74 (16.9)	Ref.	
≥50	165 (80.1)	363 (83.1)	0.82 [0.54, 1.26]	0.36
**Clinical stage,** ***n*** **(%)**				
I-IIA	195 (94.7)	428 (97.9)	Ref.	
IIB-IIIA	11 (5.3)	9 (2.1)	2.98 [1.09, 6.75]	0.03
**Histological type,** ***n*** **(%)**				
Other	11 (5.3)	37 (8.5)	Ref.	
Invasive ductal (NOS)	178 (86.4)	379 (86.7)	1.58 [0.81, 3.31]	0.20
Invasive lobular	17 (8.3)	21 (4.8)	2.72 [1.09, 7.06]	0.03
**Histological grade,** ***n*** **(%)**				
1	24 (11.7)	66 (15.1)	Ref.	
2	152 (73.8)	301 (68.9)	1.39 [0.85, 2.34]	0.20
3	30 (14.6)	70 (16.0)	1.18 [0.62, 2.23]	0.61
**LVI,** ***n*** **(%)**				
Negative	90 (43.7)	320 (73.2)	Ref.	
Positive	102 (49.5)	68 (15.6)	5.33 [3.64, 7.88]	<0.01
**Molecular subtype,** ***n*** **(%)**				
Luminal A	133 (64.6)	243 (55.6)	Ref.	
Luminal B HER2-	57 (27.7)	144 (33.0)	0.71 [0.48, 1.05]	0.08
Luminal B HER2+	7 (3.4)	22 (5.0)	0.57 [0.20, 1.42]	0.22
HER2+	5 (2.4)	7 (1.6)	1.44 [0.41, 4.90]	0.65
Triple-negative	4 (1.9)	21 (4.8)	0.36 [0.10, 1.00]	0.06

**Abbreviations:** LVI: lymphovascular invasion; OR: odds ratio; CI: confidence interval.

**Table 4 cancers-17-03396-t004:** Characterization of tumor recurrences by the management of sentinel lymph node biopsy (SLNB).

Type of Recurrence ^†^	Total	Upfront SLNB	Post-NACT SLNB
Local	5 (8.33)	5 (10.2)	0 (0.00)
Regional	9 (15.0)	8 (16.3)	1 (9.09)
Distant	34 (56.7)	26 (53.1)	8 (72.7)
Mixed	12 (20.0)	10 (20.4)	2 (18.2)
*Local* *+ Regional*	*3 (25.0)*	*2 (20.0)*	*1 (50.0)*
*Local + Distant*	*4 (33.3)*	*3 (30.0)*	*1 (50.0)*
*Regional + Distant*	*3 (33.0)*	*3 (30.0)*	*0 (0.00)*
*Local + Regional + Distant*	*2 (16.7)*	*2 (20.0)*	*0 (0.00)*
**Localization**			
**Regional, ^‡^** ***n*** **(%)**	**17 (22.9)**	**15 (88.2)**	**2 (11.7)**
**Site**			
*Axillary*	12 (70.6)	11 (73.3)	1 (50.0)
*Supraclavicular*	2 (11.7)	2 (13.3)	0 (0.00)
*Infraclavicular*	1 (5.88)	0 (0.00)	1 (50.0)
*Axillary + Supraclavicular*	1 (5.88)	1 (6.67)	0 (0.00)
*Supra and infraclavicular*	1 (5.88)	1 (6.67)	0 (0.00)
**Distant, ^‡^** ***n*** **(%)**	**43 (58.1)**	**34 (79.1)**	**9 (20.9)**
**Site**			
*Bone*	8 (18.6)	8 (23.5)	0 (0.00)
*Lung*	3 (6.98)	0 (0.00)	3 (33.3)
*Liver*	3 (6.98)	1 (2.94)	2 (22.2)
*Distant lymph nodes*	2 (4.65)	2 (5.88)	0 (0.00)
*Bone + lung*	6 (13.9)	5 (14.7)	1 (11.1)
*Bone + liver*	4 (9.30)	4 (11.7)	0 (0.00)
*Bone + distant lymph nodes*	4 (9.30)	3 (8.82)	1 (11.1)
*Bone + pleura*	2 (4.65)	2 (5.88)	0 (0.00)
*Lung + brain*	1 (2.33)	1 (2.94)	0 (0.00)
*Lung + pleura*	1 (2.33)	0 (0.00)	1 (11.1)
*Lung + distant lymph nodes*	1 (2.33)	1 (2.94)	0 (0.00)
*3 or more ^¶^*	8 (18.6)	7 (20.6)	1 (11.1)
**Treatments used,** ^†^ ***n*** **(%)**			
*ALND*	10 (16.7)	9 (90.0)	1 (10.0)
*Chemotherapy*	14 (23.3)	10 (71.4)	4 (28.6)
*Radiotherapy*	6 (10.0)	6 (100)	0 (0.00)
*Anti-HER2*	4 (6.67)	1 (25.0)	3 (75.0)
*Fulvestrant*	8 (13.3)	8 (100)	0 (0.00)
*AI*	11 (18.3)	9 (81.8)	2 (18.2)
*Endocrine therapy*	14 (23.3)	13 (92.8)	1 (7.14)
*Holobrain/bone radiation therapy*	14 (23.3)	12 (85.7)	2 (14.3)

**Abbreviations:** SLNB: Sentinel lymph node biopsy; NACT: neoadjuvant chemotherapy; ALND: axillary lymph node dissection; AI: aromatase inhibitor; †: Percentage over total number of patients with recurrence (*n* = 60); ‡: Taking into account that one patient may have had mixed recurrence, the percentage is calculated over the total number of recurrence sites (*n*’ = 74); ¶: It includes: Bone + Lung + Brain (*n* = 2), Bone + Others (*n* = 1), Liver + Pleura + Others (*n* = 1), Bone + Lung + Liver (*n* = 1), Lung + Liver + Others (*n* = 1), Bone + Lung + Liver + distant lymph nodes (*n* = 1); Bone + Lung + Liver + distant lymph nodes + skin (*n* = 1).

**Table 5 cancers-17-03396-t005:** Cox proportional hazards model estimates for time to recurrence of all types (TR) and overall survival (OS). Unadjusted analysis.

Characteristic	TR		OS	
	HR [95% CI]	*p*-Value	HR [95% CI]	*p*-Value
**Age (years)**				
<50	Reference		Reference	
≥50	0.71 [0.39, 1.28]	0.25	1.51 [0.75, 3.05]	0.25
**Tumor size**				
T0	Reference		Reference	
T1a-T1b	0.67 [0.17, 2.71]	0.57	0.42 [0.12, 1.48]	0.17
T1c	0.78 [0.26, 2.36]	0.66	0.68 [0.27, 1.69]	0.41
T2	1.81 [0.64, 5.11]	0.26	1.09 [0.46, 2.61]	0.84
T3	2.16 [0.48, 9.66]	0.31	1.57 [0.44, 5.57]	0.49
**Histological grade**				
1	Reference		Reference	
2	5.63 [1.36, 23.3]	0.02	2.80 [1.11, 7.07]	0.03
3	7.20 [1.64, 31.6]	<0.01	3.32 [1.22, 9.10]	0.02
**Involved lymph nodes**				
Negative	Reference		Reference	
1–3	0.83 [0.44, 1.55]	0.56	0.66 [0.36, 1.21]	0.18
≥4	2.21 [0.93, 5.21]	0.07	0.89 [0.28, 2.85]	0.84
**Molecular subtype**				
Luminal A	Reference		Reference	
Luminal B HER2-	3.29 [1.80, 6.02]	<0.01	1.45 [0.84, 2.51]	0.18
Luminal B HER2+	1.28 [0.37, 4.40]	0.69	0.42 [0.10, 1.78]	0.24
HER2+	1.80 [0.24, 13.6]	0.57	2.55 [0.61, 10.8]	0.20
Triple-negative	4.73 [2.02, 11.1]	<0.01	3.87 [1.92, 7.79]	<0.01
**Initial treatment**				
Surgery	Reference		Reference	
NACT	1.45 [0.73, 2.90]	0.29	1.04 [0.47, 2.29]	0.93
**Axillary surgery**				
SLNB only	Reference		Reference	
SLNB + ALND	1.22 [0.63, 2.35]	0.55	0.88 [0.45, 1.73]	0.71
**Breast surgery**				
Breast-conserving surgery	Reference		Reference	
Mastectomy	1.71 [1.00, 2.95]	0.05	2.40 [1.47, 3.91]	<0.01

**Abbreviations:** SLNB: Sentinel lymph node biopsy; ALND: axillary lymph node dissection; HR: hazard ratio; CI: confidence interval.

## Data Availability

Data are available on request from the authors.

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
