# Peer review of "Oncological Outcomes of De-Escalation of Axillary Surgery in Breast Cancer Patients at a Referral Cancer Center in Colombia"

_cancers, 2025, doi:10.3390/cancers17213396_

Round 1

Reviewer 1 Report

Comments and Suggestions for Authors

 The study provides real-world data from a middle-income country (Colombia), addressing a critical gap in the literature—most landmark trials on axillary de-escalation (e.g., ACOSOG Z0011, AMAROS) are from high-income settings. This makes the findings applicable to global oncology practice, especially in resource-limited contexts.  There are several issues need to be answered:

  1. In the post-NACT SLNB + ALND cohort, there are only 19 patients had positive SLNs in the post-NACT group, and only 5 patients underwent ALND, which limits the study’s ability to assess whether omitting ALND in Post-NACT patients with positive SLNs is safe.
  2. The prognosis analysis of this paper is not appropriate due to lots of confounding factors. PSM analysis can be used to further improve this point.

Author Response

Comments and Suggestions for Authors:

Reviewer 1

  1. In the post-NACT SLNB + ALND cohort, there are only 19 patients had positive SLNs in the post-NACT group, and only 5 patients underwent ALND, which limits the study’s ability to assess whether omitting ALND in Post-NACT patients with positive SLNs is safe.

A:

Of the 144 patients in this group, only 15 patients were N1c, which is a very low number, and of the 19 patients with positive NS, all were taken to lymphadenectomy, none of them underwent ALND omission, but other positive nodes were found in 6 of them. At the time of the unit, no lymphadenectomy was omitted in patients with positive NS after neoadjuvant chemotherapy. The text states; Again, all 19 were taken to lymphadenectomy.

  1. The prognosis analysis of this paper is not appropriate due to lots of confounding factors. PSM analysis can be used to further improve this point.

A:

The purpose of the study is to characterize the different approaches related to axillary strategies and initial treatment in patients with locally advanced breast cancer, as well as their potential impact on oncological outcomes. Although the propensity score (PS) is a fundamental statistical tool in observational studies, designed to reduce confounding bias and emulate the conditions of a randomized trial, the diversity of treatment approaches in this cohort and the small size of some groups may cause the PS model estimation to be unstable and imprecise, resulting in biased estimates. In this context, we consider alternative or complementary methods for confounding control, such as multivariable regression models. Our findings reinforce the idea that tumor biology and histopathological variables are the main determinants of oncological prognosis and disease progression, even in cases with a low axillary lymph node burden

Reviewer 2 Report

Comments and Suggestions for Authors

Comments and Suggestions for Authors:

Major comments (substantive, action-oriented)

  1. Clarify a potential inconsistency in the adjusted model for SLN positivity.
    In the multivariable logistic regression you report lobular histology with OR ~3.28; 95% CI 1.09–10.60; p=0.098 and call it “significantly associated.” Either the p-value or CI looks misstated (CI excluding 1 suggests p<0.05). Please correct and ensure text, tables, and figures are fully consistent.
  2. “N1c” vs. cN1 terminology and clip policy.
    In Results you reference “15 patients with N1c” and later state none had clip marking, while Methods say inclusion for Post-NACT SLNB requires clip placement for N1 cases. Please (a) confirm whether “N1c” is a typographical inversion of cN1; (b) reconcile the clip requirement with the fact that none of those 15 had clips; and (c) specify how target node localization was handled when clips were absent. This is important for external validity and reproducibility.
  3. Table denominators and cohort size.
    In Table 2 the footnote says “Percentage over the entire cohort (n=801)” while the analytic cohort is n=787. Please correct denominators and re-audit percentages across tables/figures.
  4. Adjuvant radiotherapy details when ALND was omitted.
    Your institutional policy for ALND omission cites high tangents/whole breast (or chest wall) plus lower axilla. Please quantify adherence (e.g., proportion receiving high tangents or RNI, dose/fractionation actually delivered) and state planning standards (fields, CTVs) so readers can judge whether nodal coverage was comparable across groups. This strengthens the conclusion that outcomes do not differ by axillary strategy.
  5. Follow-up sufficiency by subtype and subgroup.
    You note only ~42% of Post-NACT patients reached 60 months; luminal subtypes need longer follow-up for late events. Consider: (i) reporting median follow-up separately for each subgroup and subtype; (ii) a landmark or restricted mean survival time sensitivity analysis; (iii) explicitly cautioning interpretation for luminal cohorts.
  6. Handling of missing data.
    Table 1 shows non-trivial missingness (e.g., LVI “No data” ~9%). Please specify the missing-data mechanism assumed, how missingness was handled (complete case vs. imputation), and add a sensitivity analysis if feasible (e.g., include a “missing” indicator or multiple imputation for variables used in multivariable models).
  7. Events-per-variable and model diagnostics.
    Provide events counts for TR and OS, and confirm that your Cox models meet proportional hazards assumptions (e.g., Schoenfeld tests/plots). For the logistic model predicting SLN positivity, report EPV and consider penalized methods if EPV is borderline. Add a brief note on multicollinearity checks (e.g., VIFs).
  8. Explicit objective statement.
    Close the Introduction with a single-sentence, testable primary objective (and secondary objectives, if any). That will help readers align your methods and outcomes with the study aim.

Minor comments (clarifications & polish)

  • Terminology consistency. Standardize cN1/ycN0/ypN0 capitalization and spacing; ensure consistent use of Upfront SLNB vs. primary-surgery SLNB throughout.
  • Define abbreviations at first use in text and figure captions (e.g., LVI, ETD/ECE, RCB). A few are defined later or only in the Abbreviations list.
  • Quantify number of SLNs retrieved per patient by subgroup (median, IQR) in a short table or footnote; you note that 44.4% had only one node, which may affect detection—worth highlighting next to positivity rates.
  • State how RCB was computed when not reported (you mention an online tool); add a short sentence on quality control of the derived RCB (double entry or random audits).
  • Seroma and SSI rates. You report postoperative complications; consider adding confidence intervals and whether rates differed by axillary strategy (even descriptively).
  • Spell-out key radiotherapy schedules once (e.g., 5×5 Gy, 16×2.66 Gy, 25×2.0 Gy) and cite whether nodal volumes were included with each scheme in ALND-omission cases.
  • Acknowledgements/AI usage. Your ChatGPT disclosure is clear; ensure it matches Cancers’ current wording guidance.

Presentation & statistics (figures/tables and reporting)

  • Figures:
    • Flowchart (Fig. 1): Add raw numbers at each exclusion step and a final box showing analytic n for each group.
    • Kaplan–Meier panels (Fig. 3): Increase font sizes for axes/legends; ensure line styles or markers are clearly distinct in grayscale; include numbers at risk beneath each panel.
    • Forest plots: Label covariates identically to table headers; add the reference category explicitly in the caption.
  • Tables:
    • Table 1: Consider splitting into baseline and pathology/treatment characteristics for readability; keep denominators explicit where subsets are used (e.g., RCB only in NACT pts).
    • Table 2: Fix the cohort denominator typo and confirm that percentages sum as expected within each treatment category.
    • Recurrence table (Table 4): Since mixed recurrences create n’>n, add a note near the header to prevent misinterpretation; consider a simplified heat-map or stacked bars for location by strategy.
  • Effect size emphasis: Alongside p-values, consistently report HR/OR with 95% CIs (you mostly do). For clinically key comparisons (e.g., omission vs. ALND), add absolute risk differences at 5 years if feasible.
  • PH assumption & model fit: Add a brief supplement showing Schoenfeld residual tests/plots for Cox models, goodness-of-fit for logistic regression, and any influence diagnostics.

Interpretation & scope

  • Your central message—that oncologic outcomes hinge more on tumor biology (e.g., TNBC, Luminal B HER2-neg) than on the extent of axillary surgery—is well supported in multivariable analyses. Consider adding a clinically oriented paragraph in the Discussion translating how your data can operationalize decision-making (e.g., when omission is safest, pragmatic thresholds for SLN retrieval counts, LVI-positive scenarios).
  • The observed increase in Post-NACT SLNB after 2020–2021 is an important adoption signal. A short interrupted-time-series style figure (even descriptive) could make this practice shift vivid.
  • Please foreground that your regional and axillary recurrence rates lie within the range seen in trials supporting de-escalation—this helps reassure readers about generalizability in middle-income settings.

Limitations (well noted, a couple to consider expanding)

  • Selection bias toward higher axillary burden (e.g., frequent single-node retrieval; LVI prevalence) could inflate positivity—good that you discuss this; a brief propensity-style adjustment (even exploratory) for ALND vs. omission would be informative if sample permits.
  • No morbidity outcomes (lymphedema, neuropathic pain) are presented; if any data exist in clinic notes, even a small audit adds value since de-escalation is largely about morbidity reduction.

Bottom line

This is a strong, clinically relevant real-world study with careful methods and clear takeaways that reinforce the oncologic safety of axillary de-escalation when applied with appropriate selection and radiotherapy technique. Addressing the inconsistencies (p-value/CI; “N1c” vs cN1; denominator typos), expanding a bit on RT delivery details and model diagnostics, and sharpening figure/table presentation will make the manuscript even more compelling and ready for publication.

Author Response

Reviewer 2

We greatly appreciate your comments, which were very pertinent. Some points didn’t make it to the article due to space constraints, which is why we didn’t include them earlier

Major comments (substantive, action-oriented)

  1. Clarify a potential inconsistency in the adjusted model for SLN positivity.
    In the multivariable logistic regression you report lobular histology with OR ~3.28; 95% CI 1.09–10.60; p=0.098 and call it “significantly associated.” Either the p-value or CI looks misstated (CI excluding 1 suggests p<0.05). Please correct and ensure text, tables, and figures are fully consistent.

A:

We recognize that this is a typographical error. As can be seen in Figure 2, which shows the multivariate analysis, the p-value is <0.05 for lobular histology. We make the respective adjustments

  1. “N1c” vs. cN1 terminology and clip policy.
    In Results you reference “15 patients with N1c” and later state none had clip marking, while Methods say inclusion for post-NACT SLNB requires clip placement for N1 cases. Please (a) confirm whether “N1c” is a typographical inversion of cN1; (b) reconcile the clip requirement with the fact that none of those 15 had clips; and (c) specify how target node localization was handled when clips were absent. This is important for external validity and reproducibility.

A:

I agree with  your comment, the confusion of terminology is because of the translator, we corrected it.  In relation to the clip, the INC Breast Unit established management protocols for SLND and one of  these criteria was the marking of the positive node with a clip, but this was not mandatory due to the administrative problems of authorization by the HMO. and the availability of clips in the radiology service; as well as dual marking due to the availability of isosulfan blue. The other criteria are perfectly met: only patients with T1 to T3 tumors, with N0 or N1, availability of ultrasound and mammography before and after neoadjuvant chemotherapy, negative armpit by clinical signs and by breast ultrasound in the evaluation after finishing chemotherapy.  We also met the standard of resecting at least three lymph nodes in all patients  who received neoadjuvant therapy.   On the other hand , the service has extensive surgical experience, and the sentinel lymph node identification rate is almost 100%, as observed in the protocol. In patients where the sentinel lymph node is not identified, lymphadenectomy is performed, both in initial surgery and post-neoadjuvant chemotherapy. Additionally, upon reviewing axillary regional recurrence, it was remarkably low and in this study only 1   patient after neoadjuvant chemotherapy had regional recurrence

  1. Table denominators and cohort size.
    In Table 2 the footnote says “Percentage over the entire cohort (n=801)” while the analytic cohort is n=787. Please correct denominators and re-audit percentages across tables/figures.

A:

We recognize that this is a typographical error. Following the recommendations, we re-audited the percentages in the tables, figures, and text, and made the necessary adjustments.

  1. Adjuvant radiotherapy details when ALND was omitted.
    Your institutional policy for ALND omission cites high tangents/whole breast (or chest wall) plus lower axilla. Please quantify adherence (e.g., proportion receiving high tangents or RNI, dose/fractionation actually delivered) and state planning standards (fields, CTVs) so readers can judge whether nodal coverage was comparable across groups. This strengthens the conclusion that outcomes do not differ by axillary strategy.

A: You are right.  This  information is added to table 2.

What type of radiotherapy was administered?

  • Standard fractionation regimen (2 Gy × 25 fractions = 50 Gy): 55 patients (43.6%)
  • Hypofractionated regimen (2.66 Gy × 16 fractions = 42.5 Gy): 69 patients (54.7%)
  • Ultra-hypofractionated regimen (5.5 Gy × 5 fractions = 26 Gy): 2 patients (1.5%)

What radiotherapy technique was used?

  • IMRT (Intensity-Modulated Radiation Therapy): 90 patients (73.01 %)
  • 3D-CRT (Three-Dimensional Conformal Radiation Therapy): 30 patients (23.8 %)
  • IGRT (Image-Guided Radiation Therapy): 2 patient
  • VMAT (Volumetric Modulated Arc Therapy): 1 patient
  • IORT (Intraoperative Radiotherapy): 1 patient

Irradiated fields:

  • Breast – Axilla: 10 patients (7.94%)
  • Breast – Axilla – Supraclavicular fossa: 57 patients (45.24%)
  • Breast – Axilla – Supraclavicular fossa – Internal mammary chain: 5 patients (3.97%)
  • Chest wall – Axilla: 1 patient (0.79%)
  • Chest wall – Axilla – Supraclavicular fossa: 44 patients (34.92%)
  • High tangential fields were offered to 9 (7.14%) out of 126 patients

  1. Follow-up sufficiency by subtype and subgroup.
    You note only ~42% of Post-NACT patients reached 60 months; luminal subtypes need longer follow-up for late events. Consider: (i) reporting median follow-up separately for each subgroup and subtype; (ii) a landmark or restricted mean survival time sensitivity analysis; (iii) explicitly cautioning interpretation for luminal cohorts.

A:

A:

Following the reviewer's recommendations, the median follow-up time for luminal versus non-luminal subtypes was included, taking into account the type of initial treatment. Thus, the median follow-up time for patients with luminal molecular subtypes undergoing Post-NACT SLNB was 30.5 months (IQR: 15.8–55.1), while for patients with Upfront SLNB it was 57.0 months (IQR: 22.9–88.6). We acknowledge the limitation regarding the follow-up time for late events in patients with luminal subtypes, which is why this was included in the strengths and limitations section. Additionally, following the reviewer's suggestions, an additional analysis was performed using Restricted Mean Survival Time (RMST), comparing luminal versus non-luminal subtypes overall, with the following results:

Time to recurrence

Overall survival

In this case, the difference in RMST for the time-to-death outcome was 5.47 months (95% CI: 1.79–9.14). The point estimate indicated that, on average, patients with luminal subtypes survived 5.47 months longer than those in the non-luminal group after a minimum follow-up of 60 months, with this difference being statistically significant (p < 0.01).

On the other hand, the difference in RMST for time to relapse was 2.77 months (95% CI: 0.74–6.28), indicating that, on average, patients with luminal subtypes survived 2.77 months longer than the non-luminal group after at least 60 months of follow-up. However, this difference was not statistically significant (p = 0.123), suggesting that longer follow-up is necessary for patients with luminal molecular subtypes.

  1. Handling of missing data.
    Table 1 shows non-trivial missingness (e.g., LVI “No data” ~9%). Please specify the missing-data mechanism assumed, how missingness was handled (complete case vs. imputation), and add a sensitivity analysis if feasible (e.g., include a “missing” indicator or multiple imputation for variables used in multivariable models).

A:

The Functional Unit for Breast and Soft Tissue Tumors of the Instituto Nacional de Cancerología (UFM-INC) keeps a rigorous record of the information of the patients who are treated. However, there is missing information on some variables of this cohort, such as lymphovascular invasion, mainly because, given the study period, pathologists initially did not report LVI. This results in accidental loss of records, incomplete medical records, and difficulties in data collection. Considering this, the missing data mechanism was classified as "Missing Completely at Random" (MCAR) and therefore, we do not consider imputing the missing data and only work with complete cases.

  1. Events-per-variable and model diagnostics.
    Provide events counts for TR and OS and confirm that your Cox models meet proportional hazards assumptions (e.g., Schoenfeld tests/plots). For the logistic model predicting SLN positivity, report EPV and consider penalized methods if EPV is borderline. Add a brief note on multicollinearity checks (e.g., VIFs).

A:

As shown in Figure 4, the number of events per category is included in each forest plot of the Cox regression models adjusted for each outcome (column "Events"). The proportional hazards assumption was evaluated through graphical analysis of Schoenfeld residuals and the Schoenfeld test. From the visual inspection, no time-related pattern was observed for any variable included in the models. Additionally, the p-value of the test was greater than 0.05, supporting the proportional hazards hypothesis and suggesting that the assumption is satisfactorily met. Due to the limited number of figures allowed in the article, these analyses were not presented. However, the manuscript includes the p-values of the Schoenfeld statistical tests to ensure greater transparency in the results. The graphical inspection is presented below.

For Time to recurrence (TR):

For overall survival (OS):

On the other hand, for the logistic model predicting SLN positivity, the number of EPV (interpreted as ‘Events Per Variable’) by variable category is presented in Table 4, where it is observed that the majority of the variables account for ≥ 10 events. Although the categories of the molecular subtype variable did not reach an EPV ≥ 10, the goodness-of-fit measures indicated an adequate fit. Specifically, the Hosmer-Lemeshow test yielded a p-value of 0.984, confirming that the model fits the data well. On the other hand, the likelihood ratio test based on deviance showed a p-value less than 0.01, demonstrating that the model including the molecular subtype variable provides a significantly better fit compared to the model without it. Additionally, and following the recommendations, we acknowledge the importance of assessing multicollinearity and including the Variance Inflation Factor (VIF) values. Below are the VIF values for the variables included in the model:

Age

Clinical stage

Histological type

Histological grade

Lymphovascular invasion

Molecular subtype

1.05

1.04

1.09

1.37

1.07

1.48

As can be observed, these VIF values are greater than 1, suggesting a mild correlation but generally not severe enough to represent a multicollinearity problem.

  1. Explicit objective statement.
    Close the Introduction with a single-sentence, testable primary objective (and secondary objectives, if any). That will help readers align your methods and outcomes with the study aim.

A:

You are right, I correct it in the text

The primary outcome of this study was to evaluate the time to recurrence (TR), and the secondary objective was to assess overall survival. Additionally, prognostic factor s associated with positive SLND and prognostic factors associated with oncological outcomes of SLNB at the Functional Unit for Breast and Soft Tissue Tumors of the Instituto Nacional de Cancerología (UFM-INC) in Bogotá, Colombia, over a 10-year period (2013–2023). It focused on patients with early-stage I and IIA tumors who underwent SLNB as their initial surgical procedure (Upfront SLNB) and on those with locally advanced IIB and IIIA tumors (T3N1 only) treated with neoadjuvant chemotherapy, who then underwent SLNB (Post-NACT SLNB).

Minor comments (clarifications & polish)

  • Terminology consistency. Standardize cN1/ycN0/ypN0 capitalization and spacing; ensure consistent use of Upfront SLNB vs. primary-surgery SLNB throughout.

A:

We corrected the terminology  

  • Define abbreviations at first use in text and figure captions (e.g., LVI, ETD/ECE, RCB). A few are defined later or only in the Abbreviations list.

A:Terminology was corrected and abbreviations were homogenized in the text

  • Quantify number of SLNs retrieved per patient by subgroup (median, IQR) in a short table or footnote; you note that 44.4% had only one node, which may affect detection—worth highlighting next to positivity rates.

A:The information about number of SLND is in the table 1 and in the text

  • State how RCB was computed when not reported (you mention an online tool); add a short sentence on quality control of the derived RCB (double entry or random audits).

A: Our institution's pathologists are all specialized in oncology and have been utilizing the MD Anderson tool for residual breast cancer burden assessment since six years, but they don’t have formal audits. They provide all necessary data to calculate the RCB score. In this study, 9 patients  RCB calculation wasn't feasible because these patients were treated before the tool's implementation.

  • Seroma and SSI rates. You report postoperative complications; consider adding confidence intervals and whether rates differed by axillary strategy (even descriptively).

A:

For more detailed information regarding complications, the relative frequencies by axillary strategy are included along with their respective confidence intervals (regardless of sentinel lymph node positivity). No statistically significant differences were found (p-value = 1).

  • Spell-out key radiotherapy schedules once (e.g., 5×5 Gy, 16×2.66 Gy, 25×2.0 Gy) and cite whether nodal volumes were included with each scheme in ALND-omission cases.

A:

Thanks for this comment.  We had the information and it will be included in the table 2.

 Details on radiotherapy administered to patients with positive SLN and omission of ALND (n=226):

Type of radiotherapy was administered:

  • Standard fractionation regimen (2 Gy × 25 fractions = 50 Gy): 55 patients (43.6%)
  • Hypofractionated regimen (2.66 Gy × 16 fractions = 42.5 Gy): 69 patients (54.7%)
  • Ultra-hypofractionated regimen (5.5 Gy × 5 fractions = 26 Gy): 2 patients (1.5%)

Radiotherapy technique used:

  • IMRT (Intensity-Modulated Radiation Therapy): 90 patients (73.01 %)
  • 3D-CRT (Three-Dimensional Conformal Radiation Therapy): 30 patients (23.8 %)
  • IGRT (Image-Guided Radiation Therapy): 2 patient
  • VMAT (Volumetric Modulated Arc Therapy): 1 patient
  • IORT (Intraoperative Radiotherapy): 1 patient

Irradiated fields:

  • Breast – Axilla: 10 patients (7.94%)
  • Breast – Axilla – Supraclavicular fossa: 57 patients (45.24%)
  • Breast – Axilla – Supraclavicular fossa – Internal mammary chain: 5 patients (3.97%)
  • Chest wall – Axilla: 1 patient (0.79%)
  • Chest wall – Axilla – Supraclavicular fossa: 44 patients (34.92%)
  • High tangential fields were offered to 9 (7.14%) out of 126 patients

  • Acknowledgements/AI usage. Your ChatGPT disclosure is clear; ensure it matches Cancers’ current wording guidance.

A:ok

Presentation & statistics (figures/tables and reporting)

  • Figures:
    • Flowchart (Fig. 1): Add raw numbers at each exclusion step and a final box showing analytic n for each group.

A: We adjusted the flowchat for each group

    • Kaplan–Meier panels (Fig. 3): Increase font sizes for axes/legends; ensure line styles or markers are clearly distinct in grayscale; include numbers at risk beneath each panel.

A:

The Kaplan-Meier curves were adjusted according to the reviewer's recommendations. However, the grayscale aspect was not considered due to the number of categories of the molecular subtype variable.

    • Forest plots: Label covariates identically to table headers; add the reference category explicitly in the caption.

A:

The Forest plots were adjusted according to the reviewer's recommendations.

  • Tables:
    • Table 1: Consider splitting into baseline and pathology/treatment characteristics for readability; keep denominators explicit where subsets are used (e.g., RCB only in NACT pts).

A:

We believe that including all the information related to the characteristics clinical and histopathological and prognostic data of the surgical specimen is more informative, especially given the limited number of tables and figures. Therefore, we proceed with our proposal.

    • Table 2: Fix the cohort denominator typo and confirm that percentages sum as expected within each treatment category.

A:

We re-audited the percentages in the tables, figures, and text, and made the necessary adjustments.

    • Recurrence table (Table 4): Since mixed recurrences create n’>n, add a note near the header to prevent misinterpretation; consider a simplified heat-map or stacked bars for location by strategy.

A:

The table contains a footnote that states ‘Taking into account that one patient may have had mixed recurrence, the percentage is calculated over the total number of recurrence sites (n’=74)’. Although the heatmap is suitable for representing multiple recurrences, as a visualization tool it is more useful for identifying general patterns rather than for detailed analyses or precise interpretations of specific recurrences.

  • Effect size emphasis: Alongside p-values, consistently report HR/OR with 95% CIs (you mostly do). For clinically key comparisons (e.g., omission vs. ALND), add absolute risk differences at 5 years if feasible.

A:

We make adjustments according to the reviewer's recommendations

  • PH assumption & model fit: Add a brief supplement showing Schoenfeld residual tests/plots for Cox models, goodness-of-fit for logistic regression, and any influence diagnostics.

A:

We propose including the Schoenfeld residuals analysis as supplementary material.

Interpretation & scope

  • Your central message—that oncologic outcomes hinge more on tumor biology (e.g., TNBC, Luminal B HER2-neg) than on the extent of axillary surgery—is well supported in multivariable analyses. Consider adding a clinically oriented paragraph in the Discussion translating how your data can operationalize decision-making (e.g., when omission is safest, pragmatic thresholds for SLN retrieval counts, LVI-positive scenarios).
  • The observed increase in Post-NACT SLNB after 2020–2021 is an important adoption signal. A short interrupted-time-series style figure (even descriptive) could make this practice shift vivid.

A:

Initially, we considered including a figure depicting the temporal evolution of Post-NACT SLNB and Upfront SLNB treatments. However, due to limitations on the number of figures and tables, we decided not to include it in the main text. In response to the reviewer’s suggestion, this figure has now been added as supplementary material to enhance the results, illustrating the Functional Unit for Breast and Soft Tissue Tumors, Instituto Nacional de Cancerología experience with the therapeutic approaches used for these patients in the study period.

  • Please foreground that your regional and axillary recurrence rates lie within the range seen in trials supporting de-escalation—this helps reassure readers about generalizability in middle-income settings.

Limitations (well noted, a couple to consider expanding)

  • Selection bias toward higher axillary burden (e.g., frequent single-node retrieval; LVI prevalence) could inflate positivity—good that you discuss this; a brief propensity-style adjustment (even exploratory) for ALND vs. omission would be informative if sample permits.

A:

The purpose of the study is to characterize the different approaches related to axillary strategies and initial treatment in patients with locally advanced breast cancer, as well as their impact on oncological outcomes. Although the propensity score (PS) is a fundamental statistical tool in observational studies, designed to reduce confounding bias and emulate the conditions of a randomized trial, the diversity of treatment approaches in this cohort and the small size of some groups may cause the PS model estimation to be unstable and imprecise, resulting in biased estimates. In this context, we consider alternative or complementary methods for confounding control, such as multivariable regression models. Our findings reinforce the idea that tumor biology and histopathological variables are the main determinants of oncological prognosis and disease progression, even in cases with a low axillary lymph node burden

  • No morbidity outcomes (lymphedema, neuropathic pain) are presented; if any data exist in clinic notes, even a small audit adds value since de-escalation is largely about morbidity reduction.

A:

We are very honest with the information and given that this is a retrospective study, it’s very difficult to find specific information about lymphedema

Bottom line

This is a strong, clinically relevant real-world study with careful methods and clear takeaways that reinforce the oncologic safety of axillary de-escalation when applied with appropriate selection and radiotherapy technique. Addressing the inconsistencies (p-value/CI; “N1c” vs cN1; denominator typos), expanding a bit on RT delivery details and model diagnostics, and sharpening figure/table presentation will make the manuscript even more compelling and ready for publication.

We added this information in the table 2.

Reviewer 3 Report

Comments and Suggestions for Authors

Dear Authors, 

This is an important and timely study addressing the de-escalation of axillary surgery in breast cancer patients, with a large single-center cohort from Colombia over a decade. The manuscript is well written, and the findings are highly relevant for real-world practice in low- and middle-income country (LMIC) settings. The study demonstrates that sentinel lymph node biopsy (SLNB), both upfront and after neoadjuvant chemotherapy (NACT), is safe and avoids unnecessary axillary lymph node dissection (ALND) without compromising regional control or survival.

Several points could further strengthen the manuscript:

  1. Study design and confounding: Please emphasize in the discussion that the retrospective design and non-randomized assignment introduce risks of confounding by indication. Patient stage and biology differed between groups, limiting direct comparisons.

  2. Follow-up duration: Median follow-up is reasonable overall, but only ~42% of Post-NACT patients reached 60 months. This is a limitation, particularly for hormone receptor–positive tumors with later recurrence patterns, and should be highlighted more clearly.

  3. SLNB technique adherence: The methods state dual tracer and ≥3 node retrieval were protocol standards, yet the results show dual tracer was used in only 2.9% overall and ≥3 nodes retrieved in 25.7%. Please clarify whether these quality safeguards were consistently applied in the Post-NACT subgroup, where they are most critical.

  4. Targeted axillary dissection: None of the cN1 Post-NACT patients had clip marking, which is now considered standard to reduce false-negative rates. This should be acknowledged as a limitation when interpreting the safety of Post-NACT SLNB.

  5. Statistical analysis: The number of recurrences and deaths is relatively small, limiting power for subgroup comparisons. Please temper statements of “no difference” to “no significant difference detected,” acknowledging possible type II error. Additionally, a note on whether proportional hazards assumptions were tested for Cox models would be appropriate.

  6. Tables: Please report non-significant p-values rounded to two decimals. Significant findings with very small p-values may still be reported as is (i.e., 0.002).
  7. Radiotherapy and systemic therapy details: Since regional control may have been influenced by tangential or nodal radiotherapy and by systemic therapy era, more detail on radiotherapy fields/doses and systemic treatment regimens across the study period would improve interpretability.

  8. Pathology detail: Extracapsular extension (ECE) and detailed pathology quality assurance (e.g., handling of micrometastases, isolated tumor cells) were part of your protocol but are not described in the results. Providing more detail here would help readers compare your outcomes with landmark trials.

  9. Ethical approval: Please state the IRB and protocol number of the approved protocol in the corresponding section.

Author Response

Reviewer 3

We greatly appreciate your comments, which were very pertinent. Some points didn’t make it to the article due to space constraints, which is why we didn’t include them earlier.

  1. Study design and confounding: Please emphasize in the discussion that the retrospective design and non-randomized assignment introduce risks of confounding by indication. Patient stage and biology differed between groups, limiting direct comparisons.

A:

This observation will be included in the study’s  limitations section.

  1. Follow-up duration: Median follow-up is reasonable overall, but only ~42% of Post-NACT patients reached 60 months. This is a limitation, particularly for hormone receptor–positive tumors with later recurrence patterns, and should be highlighted more clearly.

A:

This observation was included in the study’s limitations section

  1. The follow-up period may be too short to detect late recurrences, especially in luminal subtype tumors. The absence of data on morbidity (lymphedema and neuropathic pain) prevents a complete assessment of the functional benefits of surgical de-escalation. Additionally, the small sample size in subgroups like Post-NACT SLNB + ALND and patients with N1C reduce the statistical power to draw firm conclusions in this clinical contex SLNB technique adherence: The methods state dual tracer and ≥3 node retrieval were protocol standards, yet the results show dual tracer was used in only 2.9% overall and ≥3 nodes retrieved in 25.7%. Please clarify whether these quality safeguards were consistently applied in the Post-NACT subgroup, where they are most critical.

A:

INC Breast Unit established management protocols for SLND and one of  these criteria was the marking of the positive node with a clip, but this was not mandatory due to the administrative problems of authorization by the HMO. and the availability of clips in the radiology service; as well as dual marking due to the availability of isosulfan blue. The other criteria are perfectly met: only patients with T1 to T3 tumors, with N0 or N1, availability of ultrasound and mammography before and after neoadjuvant chemotherapy, negative armpit by clinical signs and by breast ultrasound in the evaluation after finishing chemotherapy.   We also met the standard of resecting at least three lymph nodes in all patients  who received neoadjuvant therapy.   On the other hand , the service has extensive surgical experience, and the sentinel lymph node identification rate is almost 100%, as observed in the protocol. In patients where the sentinel lymph node is not identified, lymphadenectomy is performed, both in initial surgery and post-neoadjuvant chemotherapy. Additionally, upon reviewing axillary regional recurrence, it was remarkably low and in this study only 1   patient after neoadjuvant chemotherapy had regional recurrence.

  1. Targeted axillary dissection: None of the cN1 Post-NACT patients had clip marking, which is now considered standard to reduce false-negative rates. This should be acknowledged as a limitation when interpreting the safety of Post-NACT SLNB.
  2.  

A:

In relation to the clip, the INC Breast Unit established management protocols for SLND and one of  these criteria was the marking of the positive node with a clip, but this was not mandatory due to the administrative problems of authorization by the HMO. and the availability of clips in the radiology service; as well as dual marking due to the availability of isosulfan blue. The other criteria are perfectly met: only patients with T1 to T3 tumors, with N0 or N1, availability of ultrasound and mammography before and after neoadjuvant chemotherapy, negative armpit by clinical signs and by breast ultrasound in the evaluation after finishing chemotherapy,

  1. Statistical analysis: The number of recurrences and deaths is relatively small, limiting power for subgroup comparisons. Please temper statements of “no difference” to “no significant difference detected,” acknowledging possible type II error. Additionally, a note on whether proportional hazards assumptions were tested for Cox models would be appropriate.

A:

We make adjustments according to the reviewer's recommendations. The proportional hazards assumption was evaluated for each of the models, although we acknowledge that it was not included in the article. For greater transparency in the results, the validation of this assumption through the analysis of Schoenfeld residuals is incorporated in the article.

  1. Tables: Please report non-significant p-values rounded to two decimals. Significant findings with very small p-values may still be reported as is (i.e., 0.002).

A:

We make adjustments according to the reviewer's recommendations

  1. Radiotherapy and systemic therapy details: Since regional control may have been influenced by tangential or nodal radiotherapy and by systemic therapy era, more detail on radiotherapy fields/doses and systemic treatment regimens across the study period would improve interpretability.

  1. . Regional control could have been influenced by systemic neoadjuvant and adjuvant treatment, related with chemotherapy, given like neoadjuvant treatment in 144 patients and adjuvant treatment with chemotherapy was given in 303 patients. Among the latter group, 21 patients with residual disease post-chemotherapy were treated with T-DM1 (n=3) or capecitabine (n=18). To clarify, 282 patients (43.85%) out of 643 in the Up-front SLND group received adjuvant chemotherapy. The specific chemotherapy regimens used in the cohort are outlined below and detailed in Table 2.
  2. AC (Doxorubicin + Cyclophosphamide): 31 patients

  3. AC-T (Doxorubicin + Cyclophosphamide followed by Paclitaxel or Docetaxel): 113 patients

  4. AC-TH (Doxorubicin + Cyclophosphamide followed by Paclitaxel/Docetaxel + Trastuzumab): 17 patients

  5. Capecitabine: 18 patients

  6. TC (Docetaxel + Cyclophosphamide): 55 patients

  7. T-DM1 (Trastuzumab emtansine): 3 patients

  8. Trastuzumab monotherapy: 33 patients

  9. Other chemotherapy regimens: 33 patients

On the other hand, 605 (76.87%) patients received adjuvant radiotherapy,  577 following breast-conserving surgery and 28 following mastectomy. Among the 605 irradiated patients, 225 had positive SLND. The details of adjuvant radiotherapy treatment in this group are broken down below (this information is added to Table 2).

Type of radiotherapy was administered:

  • Standard fractionation regimen (2 Gy × 25 fractions = 50 Gy): 55 patients (43.6%)
  • Hypofractionated regimen (2.66 Gy × 16 fractions = 42.5 Gy): 69 patients (54.7%)
  • Ultra-hypofractionated regimen (5.5 Gy × 5 fractions = 26 Gy): 2 patients (1.5%)

Radiotherapy technique used:

  • IMRT (Intensity-Modulated Radiation Therapy): 90 patients (73.01 %)
  • 3D-CRT (Three-Dimensional Conformal Radiation Therapy): 30 patients (23.8 %)
  • IGRT (Image-Guided Radiation Therapy): 2 patient
  • VMAT (Volumetric Modulated Arc Therapy): 1 patient
  • IORT (Intraoperative Radiotherapy): 1 patient

Irradiated fields:

  • Breast – Axilla: 10 patients (7.94%)
  • Breast – Axilla – Supraclavicular fossa: 57 patients (45.24%)
  • Breast – Axilla – Supraclavicular fossa – Internal mammary chain: 5 patients (3.97%)
  • Chest wall – Axilla: 1 patient (0.79%)
  • Chest wall – Axilla – Supraclavicular fossa: 44 patients (34.92%)
  • High tangential fields were offered to 9 (7.14%) out of 126 patients

  1. Pathology detail: Extracapsular extension (ECE) and detailed pathology quality assurance (e.g., handling of micrometastases, isolated tumor cells) were part of your protocol but are not described in the results. Providing more detail here would help readers compare your outcomes with landmark trials.

A:

In the results section, we described the relationship between macrometastases, micrometastases and isolated tumor cells and likehood of finding other positive nodes in the ALDN. We added details on extracapsular extension that were not previously. (We added this information in table 2 and the text:  “Regarding extracapsular extension, 35 (15.55%) patients  had  extracapsular extension (ECE) >2mm, of these, 13 had negative nodes on  ALND , 12 patients had between 1  and 3  lymph nodes, 8 had between 4 and 9  and 2  had more than 10 positive  lymp nodes. On the other hand, 34 (15.11%) patients had ECE <2 mm,  14 of them underwent ALND omission and the other 20 patients had positive lymph nodes in the ALND.   6 patients  with ECE  developed distance recurrence, none  of them had regional recurrence”.

.

  • Number of patients SLNB positive without extracapsular extension (ECE):156 (69%)
  • Number of patients SLNB positive with extracapsular extension (ECE) < 2 mm: 34 (15.11%)
  • Number of patients SLNB positive with extracapsular extension (ECE) >2 mm: 35 (15.55%)

  1. Ethical approval: Please state the IRB and protocol number of the approved protocol in the corresponding section

A:

It was corrected in the article

Reviewer 4 Report

Comments and Suggestions for Authors

cancers-3904957

The manuscript entitled “Oncological outcomes of de-escalation of axillary surgery in breast cancer patients at a referral cancer center in Colombia” reports and analyzes the therapeutic outcomes of de-escalated axillary surgery in breast cancer patients. The results demonstrate that sentinel lymph node biopsy approaches are both safe and effective in early- and advanced-stage breast cancer. Overall, the manuscript is well-written and supported by high-quality tables and figures. I recommend acceptance of the paper after the authors address the following minor comments:

1. In the introduction section:

a) Please update the first paragraph with the most recent global or regional statistical data on breast cancer incidence and mortality, replacing the 2022 data if newer statistics are available.

b) Consider simplifying and restructuring this section. Some descriptive content can be moved to the Discussion section. The Introduction should remain concise while clearly highlighting the central focus and rationale of this study.

2. For the Results section, please consider revising the subtitles of each subsection so that they succinctly summarize the main conclusions of the corresponding analyses. This will improve the organization and enhance readability for the readers.

3. Consider including a paragraph outlining potential future research directions. Additionally, expand on how the findings of this study could inform clinical practice and assist physicians in making evidence-based decisions regarding axillary management strategies.

Author Response

Revisor 4

We greatly appreciate your comments, which were very pertinent. Some points didn’t make it to the article due to space constraints, which is why we didn’t include them earlier.

1.a)Please update the first paragraph with the most recent global or regional statistical data on breast cancer incidence and mortality, replacing the 2022 data if newer statistics are available.

A:  We corrected  the references

b) Consider simplifying and restructuring this section. Some descriptive content can be moved to the Discussion section. The Introduction should remain concise while clearly highlighting the central focus and rationale of this study.

The introduction summarizes the main articles. We organized the last part; however, move information to the discussion is complicated due to the lengthy discussion

2. For the Results section, please consider revising the subtitles of each subsection so that they succinctly summarize the main conclusions of the corresponding analyses. This will improve the organization and enhance readability for the readers.

These adjustments were made

3. Consider including a paragraph outlining potential future research directions. Additionally, expand on how the findings of this study could inform clinical practice and assist physicians in making evidence-based decisions regarding axillary management strategies.

This adjustment was made.

Round 2

Reviewer 3 Report

Comments and Suggestions for Authors

Thank you for your revisions.